

# High resolution neodymium characterization along the Mediterranean margins and modeling of $\varepsilon_{Nd}$ distribution in the Mediterranean basins.

M. Ayache[1], J.-C. Dutay[1], T. Arsouze[2,3], S. Révillon[4], J. Beuvier[5,6], and C. Jeandel[7]

[1]Laboratoire des Sciences du Climat et de l'Environnement LSCE/IPSL, CEA-CNRS-UVSQ, Université Paris-Saclay, 91191 Gif-sur-Yvette, France.
[2]ENSTA ParisTech, Université Paris-Saclay, 828 bd des Maréchaux, 91762 Palaiseau cedex, France
[3]Laboratoire de Météorologie Dynamique, École Polytechnique, Palaiseau, France.
[4]SEDISOR/UMR6538 "Laboratoire Domaines Océaniques", IUEM, CNRS-UBO, Plouzané, France
[5]Mercator-Océan, Ramonville Saint-Agne, France.
[6]Météo-France, Toulouse, France.
[7]LEGOS, Université de Toulouse, CNRS, CNES, IRD, UPS, Toulouse, France

*Correspondence to:* M. Ayache (mohamed.ayache@lsce.ipsl.fr).

**Abstract.** An extensive compilation of published neodymium (Nd) concentrations and isotopic compositions (Nd IC) was realized in order to establish a new database and a map (using a high resolution geological map of the area) of the distribution of these parameters for all the Mediterranean margins. Data were extracted from different kinds of samples: river solid discharge deposited on the shelf, sedimentary material collected on the margin or geological material outcropping above or close to a
5 margin. Additional analyses of surface sediments were done, in order to improve this dataset in key areas (e.g., Sicilian strait).

The Mediterranean margin Nd isotopic signatures vary from non-radiogenic values around the Gulf of Lions, ($\varepsilon_{Nd}$ values $\sim$ -11) to radiogenic values around the Aegean and the Levantine sub-basins up to +6. Using a high resolution regional oceanic model (1/12° of horizontal resolution), $\varepsilon_{Nd}$ distribution was simulated for the first time in the Mediterranean Sea.

The high resolution of the model provides the unique opportunity to represent a realistic thermohaline circulation in the
10 basin and thus apprehend the processes governing the Nd isotope distribution in the marine environment. Results reinforce the preceding conclusions on boundary exchange "BE" as an important process in the Nd oceanic cycle. Nevertheless this approach simulates a slightly too radiogenic value in the Med Sea, this bias will likely be corrected once the dust and river inputs will be included in the model.

This work highlights that a significant interannual variability of $\varepsilon_{Nd}$ distribution in seawater could occur. In particular,
important hydrological events such as the Eastern Mediterranean Transient (EMT), associated with deep water formed in the Aegean sub-basin, could induce a shift in $\varepsilon_{Nd}$ at deep/intermediate depths that could be noticeable in the Eastern part of the basin. This underlines that the temporal and geographical variations of $\varepsilon_{Nd}$ could represent an interesting insight of Nd as tracer of the Mediterranean Sea circulation, in particular in the context of paleo-oceanographic applications.



# 1 Introduction

The Mediterranean Sea is a semi-enclosed sea of great interest because it is submitted to large range of dynamical processes and interactions, such as strong air-sea exchanges leading to open-sea deep water convection feeding a thermohaline circulation cell (Malanotte-Rizzoli and Robinson, 1988), strait transports and dynamics or cross-shore exchanges. From a biogeochemical perspective, it is a region receiving the highest aerosol loads owing to air masses carrying numerous and various aerosol types (Lelieveld et al., 2002; Nabat et al., 2014), where oligotrophy occurs and with a characteristic dynamic of the deep chlorophyll maximum (Mignot et al., 2014). Under the stress of the global change and anthropogenic forcing, understanding the functioning of the Mediterranean Sea and quantifying the biogeochemical cycles is a priority (MerMex-Group, 2011).

Neodymium (Nd) is a Rare Earth Element (REE) with seven naturally occurring isotopes, all stable. The radiogenic isotope $^{143}$Nd is produced by the radioactive $\alpha$-decay of $^{147}$Sm. At the continent surface, the Nd isotopic composition (usually expressed as $\varepsilon_{Nd}$[1] of a given material is a function of the Sm/Nd ratio characterizing this material, which is primarily a function of its age and lithology. On a global scale, it is higher in the Earth's mantle compared to its crust. As a consequence, the $\varepsilon_{Nd}$ of the continents presents a heterogeneous distribution (Goldstein and Hemming, 2003; Jeandel et al., 2007).

Nd residence time range from 700 to 1500 yr in the global ocean (e.g., Lacan et al., 2012), long enough to be transported within the global thermohaline circulation system and short enough to avoid complete homogenisation. Therefore, $\varepsilon_{Nd}$ is often considered to be a "quasi-conservative" tracer. In other words, $\varepsilon_{Nd}$ values of the water masses could be conserved up to long distance from the source of lithogenic inputs. In such context, it could be used to tag water masses with distinct isotopic compositions in order to constrain water mass mixing and pathways, and the thermohaline circulation in modern and paleo ocean circulation (e.g., Lacan and Jeandel, 2005; Jeandel, 1993; Jeandel et al., 1998; Frank, 2002; Goldstein and Hemming, 2003; Piotrowski et al., 2004; Stichel et al., 2012; Piotrowski et al., 2012; Martin et al., 2012; Pena et al., 2013; Molina-Kescher et al., 2014; Arsouze et al., 2008). However, because Nd is particle reactive, Nd parameters are also successfully used to study Nd exchange between dissolved and particulate phases (Bertram and Elderfield, 1993; Henry et al., 1994; Jeandel et al., 1995; Tachikawa et al., 1999; Tachikawa, 2003; Garcia-Solsona et al., 2014; Rousseau et al., 2015; Haley et al., 2014).

Nd sources to the ocean are lithogenic, and the mean $\varepsilon_{Nd}$ of an oceanic basin is representative of the surroundings continents (Jeandel et al., 2007). During the last few years, significant progress has been made in understanding how different water masses acquire their Nd IC. Early 2000s, (Tachikawa, 2003) and (Lacan and Jeandel, 2005), suggested that exchange of Nd between the sediments deposited on the oceanic margins and the waters flowing along these margins, called the "Boundary Exchange" (BE) was the missing Nd source that could balance both the concentration and isotopic distributions of Nd on regional and world's scales. Since these pioneer works, many studies confirmed this hypothesis (Rickli et al., 2009, 2010; Stichel et al., 2012; Wilson et al., 2012; Grenier et al., 2013; Carter et al., 2012). The modelling studies have reached the same conclusions on the relative importance of the BE on the Nd oceanic cycle on the global scale, although dust and river inputs could locally

---

[1]$\varepsilon_{Nd} = ((^{143}\text{Nd}/^{144}\text{Nd})_{sample}/(^{143}\text{Nd}/^{144}\text{Nd})_{CHUR}).^4 10$ , where $(^{143}\text{Nd}/^{144}\text{Nd})_{CHUR} = 0512638$ is the averaged earth value (Jacobsen and Wasserburg, 1980)



affect the surface waters, as off the Saharan for example (Arsouze et al., 2007; Siddall et al., 2008; Arsouze et al., 2009; Rempfer et al., 2011).

The Nd influx brought by the Atlantic Inflow at the Strait of Gibraltar is smaller than the Nd outflux exiting with the Mediterranean Outflow (Tachikawa et al., 2004; Greaves et al., 1991; Henry et al., 1994). Furthermore, the $\varepsilon_{Nd}$ value of the
Mediterranean Outflow ($\varepsilon_{Nd}$ = -9.5; Henry et al. (1994)) is higher than that of the Atlantic inflow ($\varepsilon_{Nd}$ =-11.8; Spivack and Wasserburg (1988)). Thus, a source of radiogenic Nd is required to balance these fluxes.

Frost et al. (1986) and Spivack and Wasserburg (1988) proposed that the additional Nd source might be the partial dissolution of river particles and/or aeolian particles. Greaves et al. (1991) argued that the missing source might be rather of marine origin. Schijf et al. (1991) suggested that the Black Sea was a net source to the Mediterranean Sea. Based on a two box model Henry
et al. (1994) suggested that the $\varepsilon_{Nd}$ in the Ligurian sub-basin deep waters required an exchange involving $30 \pm 20\%$ of the sinking particles of atmospheric origin. Finally, Tachikawa et al. (2004) proposed that the missing term could be sediments deposited on the margins. In other words, the origin of this radiogenic input remains unclear. The present study aims to compile data and develop modelling tools for clarifying this issue.

The circulation of the Mediterranean Sea is driven by the fact that the mean evaporation exceeds the mean precipitation
leading to a density increase along surface water masses paths and subsequent strong convective events in winter. The main deep water sources are located in the Gulf of Lions (south of France) for the Western Mediterranean Sea (WMed), and the Adriatic sub-basin for the Eastern Mediterranean Sea (EMed; Millot and Taupier-Letage (2005)). In the mid-1990s a shift in the deep water formation site occurred during the Eastern Mediterranean Transient (EMT) events. The EMT describes a temporary change in the Eastern Mediterranean Deep Water (EMDW) formation site that switched from the Adriatic to the
Aegean sub-basin (Roether et al., 1996, 2007; Lascaratos et al., 1999; Malanotte-Rizzoli et al., 1999; Theocharis et al., 1992, 1999) . The new source has produced large quantities of very dense water masses, in particular the Cretan Deep Water (CDW) that overflowed through the Cretan Arc Straits and subsequently filled the eastern Mediterranean with waters denser than the previously existing deep and bottom water. However, EMT was revealed using hydrographic tracers and anthropogenic ones such as CFC and $^{3}$H, both transients and not imprinted in the sediments. Establishing the occurrence of similar EMT event in
the past would require the identification of proxies that clearly identify the distribution and circulation of the different water masses and that is memorized in the sediments.

Tachikawa et al. (2004) demonstrated that the Nd isotopic signature is more conservative than the salinity in the Mediterranean Sea, the latter being strongly affected by the evaporation. In addition, these authors revealed that the Mediterranean water masses are well distinguished by their Nd isotopic signatures. The Mediterranean Sea makes an excellent "laboratory
test" basin for studying the potential $\varepsilon_{Nd}$ distribution variations as it is a semi enclosed basin with a quite short residence time of the waters (50-100 years; Millot and Taupier-Letage (2005)). This paper also aims to investigate how EMT events affect the Nd distribution in the Mediterranean basin, in order to estimate the potential of this tracer to characterize from paleo archive (e.g., corals, foraminifera) the occurrence of such event in the past. Modelling represents an appropriate tool to address this question.



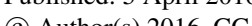


In this study, we developed a new modelling platform for simulating Nd isotopic composition at high resolution in the Mediterranean basin.

First, results of a dense compilation of the concentrations and isotopic compositions of the different materials that constitute the Mediterranean margins and are expected to interact with the water masses are presented. This high resolution mapping was established using a detailed geological map, providing the most realistic representation of the Mediterranean geology existing so far (see Fig. 1, Fig. 2, and Appendix 1).

Then, following the protocol made up for the global scale by (Arsouze et al., 2007), we implemented the neodymium in a high-resolution regional model (NEMO-MED12). We used dissolved $\varepsilon_{Nd}$ data compiled by Tachikawa et al. (2004) to evaluate the ability of this model to reproduce the main features of the circulation and mixing of the Med Sea water masses for which Nd signatures are known. These tools provided perspectives on i) the role of the BE on the distribution of $\varepsilon_{Nd}$ parameters, and ii) the impact of the inter-annual variability of the thermohaline circulation (e.g., EMT event) on the modelled $\varepsilon_{Nd}$ distribution.

## 2   Data compilation and representation on the Mediterranean margins

Here we present the data compilation procedure and resulting map (Fig. 2) allowing us to characterize the Nd isotopic signatures and concentrations of all the margins surrounding the Mediterranean Sea.

Our approach was to prioritize the use of data directly measured in the outcropping sediments or geological fields, as proposed by (Jeandel et al., 2007). An extensive compilation of all published Nd concentrations and isotopic values was made using the EarthChem database (http://www.earthchem.org) with a zoom on the Mediterranean region (latitude between 28°N to 48°N, and between 10°W and 40°E in longitude). This yielded asset of more than 14200 discrete data reported in Appendix 1 and located in Fig. 1. When data were missing in crucial areas (e.g., strait of Sicily), we directly measured them on top core sediments.

Finally, we used a high resolution numerical geological map to extrapolate data from similar geological areas. Below we briefly discuss the pros and cons of this latter approach.

### 2.1   Sediment core tops and erodable material data

Surface sediments collected on the shelf or the slope were our first choice because they provide direct information on the geochemical and isotopic characteristics of the material in contact with the water masses. Sediments deposited during the recent Holocene were taken in consideration. When surface sediment data were missing, we took into account the Nd parameters of erodable material deposited along the coasts (Jeandel et al., 2007). Earthchem database provides a good spatial covering in the northern rive (Fig. 1), contrastingly to the southern rive (e.g., Algerian coast), while there are no published data for the Tunisian, Libyan and Egyptian coasts. We therefore distinguished between the two cases sections 2.1.1 and 2.1.2.



### 2.1.1 Areas with high spatial resolution of available data

For this kind of region (i.e., Italian coast), and for each geological province, we carefully established the extent to which we could extrapolate the measured values to the whole province. The average $\varepsilon_{Nd}$ values for each geological province were calculated taken into account the number of cores collected in a given area, and the geochemical characteristics of the anal-

ysed sediments. We neglected data for specific area like very small volcanoes not representative of the geochemical and iso-topic characteristics of the region, as in the Strait of Sicily for example (see Appendix 1 reports the full treatment, results and uncertainties). This approach was generating relatively robust Nd isotopic signatures characterizing the whole Northern Mediterranean Sea margins, with standard deviation less than 2 $\varepsilon_{Nd}$ units in most of the cases.

### 2.1.2 Areas with low spatial resolution of available data

For the southern coast, we gathered sediment samples from miscellaneous origins. Those were collected during the cruises of ETNA80, DEDALE, and NOE near the Tunisian, Libyan and the Egyptian coasts respectively (see Table 1). In the Strait of Sicily samples were collected along two transects, Sciacca-Pantelleria (SP hereafter) and Pozzallo-Malta (PM hereafter), perpendicular to the southern coast of Sicily (Tranchida et al., 2011). The sampling site identifications, depths, collection dates and positions are compiled in Table 1.

Sediment samples were analysed for Nd radiogenic isotopes. About 100 mg of samples were weighted and dissolved in teflon beakers in a mixture of ultrapure quartex HF ($^{24}$N), $HN0_3$ ($^{14}$N) and $HClO_4$ ($^{24}$N) for four days at 160°C on a hot plate. After evaporation to dryness samples were dissolved in aqua regia and heated for 24 hours at 130°C. Nd fractions were chemically separated following conventional column chemistry procedures described in Révillon et al. (2011). Nd isotope compositions were measured in static mode on a Thermo TRITON at the PSO (Pole de Spectrométrie Océan) in Brest, France.

All measured Nd ratios were normalized to $^{146}$Nd/$^{144}$Nd = 0.7219. During the course of analysis, Nd standard solution La Jolla gave 0.511854 ± 0.000008 ($2\alpha$; n=28, recommended value 0.511850) and JNdi gave 0.512099 ± 0.000010 ($2\alpha$; n=6, recommended value 0.512100). Procedural blanks were all below 200 pg and therefore negligible in all case.

### 2.2 River and dust inputs

Dissolved Nd in river water is efficiently removed from solution by coagulation of colloids during the estuarine mixing. The

recent compilation of Rousseau et al. (2015) confirmed that on a global scale 71.8 ±16% of dissolved riverine Nd is removed by this process, in agreement with preceding works (e.g., Elderfield et al., 1990). In addition, these authors evidenced that lithogenic Nd is released at higher salinities by the Suspended Particulate Material discharged by the river. Globally, this could represent 5,700 ± 2,600 Mg of dissolved Nd annually brought to the ocean by this mechanism, a flux 6-17 times larger than the dissolved one, and 8 to 21 times larger than the atmospheric flux, assuming 2% dust dissolution. Note that the mechanism

evidenced in the Amazon estuary by Rousseau et al. (2015) does not describe all the processes that likely affect the sediments deposited on the shelves and margins, hypothesis modelled by Arsouze et al. (2009) and Rempfer et al. (2011). Dust input



is also difficult to constrain: while flux is sporadic and hard to characterize, establishing the fraction dissolved at the air-sea interface is also challenging.

The main river systems of the Mediterranean basin are the Nile, Po and Rhone. River plume extensions were established using maps and satellite images from data banks provided by Ludwig et al. (2009).

Nd input derived from the Nile river water and/or particles could be transported eastward and northward to the Rhodes Gyre where the LIW is formed. Tachikawa et al. (2004) suggest that the most significant radiogenic Nd source to the EMed is partially dissolved Nile River particles, radiogenic Nd supplies to the eastern basin being formed by dissolved and particulate loads ($\varepsilon_{Nd}$Nd of about $\sim$ -4).

Henry et al. (1994) have studied the potential impact of river inputs on the Nd isotopic composition of the WMed. The Rhone transports 80% of the solid riverine discharge into the northwestern Mediterranean Sea (Leveau and Coste, 1987). According to Henry et al. (1994) the Rhone dissolved water and superficial sediments display an average $\varepsilon_{Nd}$ value of -10.2 $\pm$ 0.5.

Contrastingly, the Po river is less documented. The geographical extension of Po river drainage basin was defined using the digital geological map referenced above, and Ludwig et al. (2009) data base. These information allowed us to extract the Nd isotopic signatures from the work of Conticelli et al. (2009), Prelević et al. (2008) and Owen (2007). These studies are the most representative of the Po drainage basin so far.

Finally, we included aeolian inputs to our compilation. To this extent, the review of Scheuvens et al. (2013) was of great help. Indeed, this work presents a review of bulk composition data of northern dust inputs, their potential sediment sources and their elemental, isotope and mineralogical characteristics. Actually, these aeolian data will not be immediately used in our modelling approach. However, we considered relevant to present them with the remaining data, which allowed us to propose the most comprehensive data set.

All the discrete data extracted from the literature are reported in Appendix 1 and Fig. 1.

## 2.3 Extrapolation

Extrapolating the discrete data collected is required to allocate the margins with continuous Nd concentration and isotopic compositions. In other words, attributing an isotopic signature and Nd concentration to any margin area liable to be in contact with the waters flowing through. To this end, the tools developed by Jeandel et al. (2007) for the world margins were adapted for the Mediterranean Sea.

Because the Nd isotopic composition of any field is closely related to its geological nature and age (O'Nions et al., 1979; Goldstein et al., 1984, 1997; Allegre, 2005) we used a high resolution digital geological map which provides the contours of the fields (Fig. 1) of a given geological age and type (http://www.geologie.ens.fr/spiplabocnrs/spip.php?rubrique67). This allowed us to estimate the size of the coastal segments that could provide material with the same Nd characteristics.

In poorly documented areas, we first considered the age and geochemical nature of the field (Jeandel et al., 2007). Then we applied the same isotopic signature as similar fields of the same age based on the Nd model-age relationships (Allegre, 2005; Goldstein et al., 1984, 1997; O'Nions et al., 1979). This was thoroughly done by checking the geology, geochemistry and Nd signatures of the fields identified as the source of deposited material. Note that the resolution of the available data in



the Mediterranean basin is relatively high, reducing the uncertainties of the approach described above (see section 2, Appendix 1), Fig. 2b reveals the improvement allowed by this approach by comparing the high resolution patchwork of field documented yet compared to an extraction of the Mediterranean Sea basin (Fig. 2a) from the global low resolution distribution of Jeandel et al. (2007)(see section 4.1).

## 3  Modelling the Nd isotopic composition

### 3.1  Description of the model

We use the free-surface ocean circulation model NEMO (Nucleus for European Modelling of the Ocean) (Madec and NEMO-Team., 2008) in a regional configuration called NEMO-MED12 (Beuvier et al., 2012a), already used for biogeochemical studies (Ayache et al., 2015b; Guyennon et al., 2015; Ayache et al., 2015a; Palmiéri et al., 2015). This model uses the standard ORCA grid of NEMO at 1/12° resolution. This corresponds in the Mediterranean area to a grid cell size varying with latitude between 6 and 8 km, from 46°N to 30° N and also extends into the Atlantic Ocean to 11° W (buffer zone). Vertical resolution varies with depth from $\Delta Z = 1$ m at the surface to $\Delta Z = 450$ m at the bottom with 35 levels in the first 1000 m (50 levels in total).

Daily mean fields of momentum, freshwater flux (evaporation minus precipitation) and net heat flux from the high resolution atmospheric data set (ARPERA) are used for the air-sea fluxes (Herrmann et al., 2010; Herrmann and Somot, 2008). The heat flux is applied with a retroaction term using the ERA-40 Sea Surface Temperature (SST).

The initial state (temperature, salinity) for the Mediterranean Sea come from the MedAtlas-II (Rixen et al., 2005; MEDAR-MedAtlas-group, 2002) climatology weighted by a low-pass filter with a time window of 10 years using the MedAtlas data covering the 1955-1965 period, following (Beuvier et al., 2012b). For the temperature and salinity in the buffer zone (west of Gibraltar strait), the initial state is prescribed from the 2005 World Ocean Atlas (Antonov et al., 2006; Locarnini et al., 2006). River runoff is prescribed from the interannual data set of (Ludwig et al., 2009). The Black Sea is not explicitly included in the models, but is rather treated as one of the major freshwater sources of the Mediterranean Sea located at the Dardanelles strait, with a flux corresponding to Dardanelles' net budget estimates of (Stanev and Peneva, 2002).

NEMO-MED12 model simulates the main features of the thermohaline circulation and mixing of the Mediterranean Sea water masses and their interannual variability. In particular, the propagation of the Levantine Intermediate Water (LIW) from the Eastern to the western basin is produced with realistic timescale compared to observations (Ayache et al., 2015a). However, some aspects of the model still need to be improved. For example the formation of Adriatic Deep Water (AdDW) is too weak, leading to a too low contribution to the EMDW in the Ionian sub-basin (Palmiéri et al., 2015; Ayache et al., 2015a, b). The atmospheric forcing used by Beuvier et al. (2012b) includes some modifications to improve dense water fluxes through the Cretan Arc during the EMT. As established in the previous version of NEMO-MED8 (1/8° of horizontal resolution, Beuvier et al. (2010)), the model is able to reproduce a transient deep-water formation as observed for the EMT, but the simulated transient produced less Eastern Mediterranean Deep Water (EMDW). Beuvier et al. (2012b) later performed a sensitivity test with modified forcing. The ARPERA forcing were modified over the Aegean sub-basin, by increasing daily water loss by 1.5 mm, daily surface heat loss by 40 W m$^{-2}$, and the daily wind stress modulus by 0.02 N m$^{-2}$ during November to March in





the winters of 1991-1992 and 1992-1993, as done by Herrmann and Somot (2008) to study the Gulf of Lions deep convection. This resulted in average winter time increases in heat loss (+18 %), water loss (+41 %) and wind intensity (+17 %) over the Aegean sub-basin. These changes generate an improved circulation that satisfyingly reproduced the formation and renewal of the deep water in the Eastern basin during the EMT event (Ayache et al., 2015a). These performances of the dynamical model
have to be kept in mind when analysing the Nd simulations.

## 3.2   The tracer model

Using different modelling approaches, Arsouze et al. (2007, 2010) and Rempfer et al. (2011) have shown that the exchange between the continental margins and seawater, the "boundary exchange" (BE) represents the major source of Nd on the global scale. This source represents more than 90 % of the total input, whereas dissolved riverine and dust inputs could be significant
in the upper 500 m only. However, so far only little is known about the importance of the BE in a semi-enclosed and/or interiors basins like the Mediterranean Sea, where atmospheric and river fluxes could have significant impacts on the $\varepsilon_{Nd}$ distribution too.

    We chose as a first approach to simulate only the Nd isotopic composition ($\varepsilon_{Nd}$) in order to test the BE hypothesis in the Mediterranean Sea (Arsouze et al., 2007). This approach does not require to simulate explicitly the Nd concentration, allowing us to focus on the timescale of the process studied. As in Arsouze et al. (2007, 2010), $\varepsilon_{Nd}$ is implemented in the model as
a passive conservative tracer which does not affect ocean circulation. It is transported in the Mediterranean Sea by NEMO-MED12 physical fields using a classical advection-diffusion equation, including the sources and sinks (SMS term, eq1).The rate of change of oceanic Nd isotopic composition is:

$$\frac{\delta \varepsilon_{Nd}}{\delta t} = S(\varepsilon_{Nd}) - U \cdot \nabla \varepsilon_{Nd} + \nabla \cdot (K \nabla \varepsilon_{Nd}), \tag{1}$$

where S($\varepsilon_{Nd}$) represents the SMS term, U $\cdot \nabla \varepsilon_{Nd}$ is the three-dimensional advection and $\nabla \cdot (K \nabla \varepsilon_{Nd})$ is the lateral and vertical diffusion of $\varepsilon_{Nd}$.

    Since $\varepsilon_{Nd}$ is a passive tracer, simulations could be run in off-line mode using the pre-computed transport fields (U, V, W) from the NEMO-MED12 dynamical model (Beuvier et al., 2012b). Physical forcing fields are read daily and interpolated each time step of 20 min Offline simulations are performed for computational efficiency allowing many sensitivity tests on the SMS
term parameterization. The same approach was used by Ayache et al. (2015b) to simulate the mantle and crustal helium isotope signature, by Ayache et al. (2015a) to model the anthropogenic tritium invasion, and by Palmiéri et al. (2015) to simulate CFCs and anthropogenic carbon storage.

    The only SMS term taken into account in the present study is BE (Arsouze et al., 2007, 2010). It is parameterized by a relaxing equation between the ocean and the continental margin:

$S(\varepsilon_{Nd}) = 1/\tau (\varepsilon_{Ndmargin} - \varepsilon_{Nd}).mask_{margin}, \tag{2}$



where $\tau$ is the characteristic relaxing time (i.e., the characteristic time needed to transfer isotopic properties from the continental margin to the ocean), $\varepsilon_{Nd}$ is the Nd isotopic composition of seawater, $\varepsilon_{Ndmargin}$ is the value of the material deposited along the continental margin (see section 2), and $mask_{margin}$ is the percentage of continental margin in the grid box which represents the proportion of the surface in the grid where the BE process occurs. This quantity is estimated from the high-resolution bathymetry of the $10^{th}$ version of the Mercator-LEGOS bathymetry at a resolution of 30" × 30".

The topographic extension of the oceanic margins of the Mediterranean Sea has been chosen to the ∼540 m (Fig. 3) following the margin definition used to model the iron cycle in the Mediterranean Sea by Palmiéri (2014).

The exchanges of the Nd with the Atlantic Ocean are specified through a buffer zone between 11W° and the Strait of Gibraltar. $\varepsilon_{Nd}$ values in the buffer zone are prescribed from observation using NE Atl. MED-15 vertical profile from Lacan et al. (2012).

We established some sensitivity experiments regarding the optimal value of $\tau$ in the Med basin (Arsouze et al., 2007, 2010). In this aim, six tests were performed referred as EXP1, EXP2, EXP3 and EXP4 with $\tau$ =1 month, 3 months, 6 months, and 1 year, respectively. As surface ocean currents are generally more dynamic than deep ones, providing more energy for sediment-seawater interactions, we realized an additional simulation (EXP5) wherein $\tau$ increases exponentially with depth from 1 month at the surface to 1 year at 600 m depth. We also explored the possibility that our BE parameterization might be dependent on the mineralogical maturity of margin sediments (e.g., granitic vs. basaltic). Hence, relaxing time $\tau$ in EXP6 is varying linearly on a timescale of 1 month for the most radiogenic isotopic signature (i.e., $\varepsilon_{Nd}$ = +6 on the extreme east of the Mediterranean margin) to 1 years for the most non-radiogenic values (i.e., $\varepsilon_{Nd}$ =-12 along Spanish coast).

The simulations were initialized with uniform isotopic composition of $\varepsilon_{Nd}$=-7 and integrated to steady state, i.e., the global averaged drift was less than $10^{-3}$ $\varepsilon_{Nd}$ per thousand years, for more than 75 years of spin-up run.

## 4 Results

### 4.1 Map of the outcropping Nd values

Results of the Nd parameter mapping are represented in Fig. 2, cold colours represent the old non-radiogenic rocks whereas the warm colours correspond to the recent radiogenic ones.

Tectonic and associated volcanic activities led to the very complex morphology and geology in the Mediterranean region, comprising small islands (e.g., Corsica, Cyprus), sub-basins (e.g., Adriatic, Aegean, and Tyrrhenian), and many narrow straits (e.g., Sicily channel, Otranto Passage). This particular context prevents the use of low resolution grid to represent properly this region. This motivated the realisation of the high resolution (1/12°x1/12°) version of the Nd isotopic signature (Fig. 2b) and Nd concentration (see Appendix. 5) for this basin.

The general trend is that the margin Nd isotopic signatures vary from non-radiogenic values in the WMed, to radiogenic values when reaching the Aegean and Egyptians coasts, the most radiogenic fields ($\varepsilon_{Nd}$ up to +6) being located around the east border of the Levantine sub-basins, and in the volcanic region of the south of Italy (Fig. 2b). Contrastingly, the south Sicilian fields and the north Alboran sub-basin are characterized by the most negative isotopic signature (values around -12).



The Algerian, Tunisian, French and Spanish coasts display relatively homogeneous values between -11.5 and -10. Such West-East gradient of Nd isotopic signature is also observed in the seawater data, where poorly radiogenic waters from the Atlantic are progressively shifted toward more radiogenic values in the Levantine basin (Tachikawa et al., 2004).

All the details revealed by this new high resolution map will be used to set boundary conditions in the regional simulation
(see section 3, Fig. 3).

### 4.2   The characteristic margin-to-ocean exchange time.

We first explored the impact of changing the value of the relaxing time on the $\varepsilon_{Nd}$ distribution in the Med Sea. This was made following the strategy adopted by (Arsouze et al., 2010) in the North Atlantic basin, although NEMO-MED12 model has higher horizontal and vertical resolutions ($1/12°$ in this study compared to $1/4°$ in Arsouze et al. (2010)).

The results of these different sensitivity tests are compared with in-situ observations collected by Tachikawa et al. (2004) using correlation plots, coloured maps and sections (Fig. 4 and 5).

The simulated $\varepsilon_{Nd}$ distributions in EXP2 and EXP3 (relaxing time of 3 and 6 months respectively) present the best correlation with in-situ data relative to the other experiments, with correlation coefficients close to 0.71 and 0.61 respectively (Tab. 3). The difference between in-situ data (dashed line) and the different sensitivity experiments as a function of depth
(Fig. 4b) reveals that EXP2 provides the best agreement with observations, despite a slight underestimation of $\varepsilon_{Nd}$ between 0.3 and 1 $\varepsilon_{Nd}$ units. The data-model differences are more important for the other experiments which produced too radiogenic simulations (of more than 2 $\varepsilon_{Nd}$ units). The horizontal distribution of $\varepsilon_{Nd}$ (Appendix 2, 3 and 4) confirms this statistical correlation, showing that only EXP2 and EXP3 produced a reasonable East-West gradient of $\varepsilon_{Nd}$. EXP1 generated too pronounced $\varepsilon_{Nd}$ geographical gradients particularly in surface waters along the continental margins suggesting an overestimation
of the exchange compared to the transport. On the opposite a simulation with a relaxing time of 1 year (EXP4) leads to an homogeneous $\varepsilon_{Nd}$ distribution in surface and deep waters with a low data-model correlation, indicating an underestimation of the boundary exchange process. EXP5 ($\tau$ increases with depth) conducted to a strong gradient over the entire water column in WMed, showing that surface to deep variation of the BE rate was likely overestimated in this simulation. EXP6 displayed a realistic E-W gradient in the surface waters, but too homogeneous $\varepsilon_{Nd}$ signal in the intermediate and deep waters, suggesting
that the BE rate seems weakly affected by the lithology of the margin sediments. Finally, we consider that the characteristic exchange time providing the best agreement with observations is close to 3 months. This value is consistent with the results obtained by Arsouze et al. (2010) with its simulation of the North Atlantic area. Therefore we will only consider EXP2 for the rest of our analysis.

### 4.3   The $\varepsilon_{Nd}$ distribution

The monthly-averaged $\varepsilon_{Nd}$ horizontal distributions resulting from EXP2 for the surface waters (0-200 m), the intermediate waters (200 - 600 m) and the deep waters (600 - 3500 m), are represented in Fig. 5a, 5b and 5c respectively, together with the data from Tachikawa et al. (2004). The model results are extracted after the steady state (in 1987) of the simulation and considered as representative of pre-EMT situation (i.e., EMT, Roether et al. (2007)).





The model correctly simulates the pronounced $\varepsilon_{Nd}$ E-W gradient characterizing the surface waters (Fig. 5a). The values simulated in the WMED and Eastern Levantine basin are consistent with the observations while the simulated values in the Aegean and central Levantine basin tend to be too radiogenic. At intermediate depths, both modelled and observed E-W gradients are less pronounced than at the surface (Fig. 5b). However averaged simulated values are relatively too radiogenic at

5 the intermediate level (-5.8 compared to -8.7±1.9, Tab.2). Especially high $\varepsilon_{Nd}$ signatures are simulated over the Sicily channel and the Tyrrhenian sub-basins (Fig. 5b), but the lack of observations prevent us to assess their consistency. The most radiogenic simulated values are found in the north Aegean sub-basin and largely overestimate the observations. The $\varepsilon_{Nd}$ distribution in the deep waters is relatively homogeneous over the whole basin except in the Aegean sub-basin and Sicilian channel (Fig. 5c).

Levantine Intermediate water mass (LIW) is well identified by its marked radiogenic signature. LIW is produced in the

10 Levantine sub-basin before passing Crete at 28 °E, where measured $\varepsilon_{Nd}$ values reach -5 (Fig. 5). The $\varepsilon_{Nd}$ isotopic signature is well identified over the entire LIW trajectory at the intermediate level (between 200 and 600 m depth), with values around -4.8 in the Algerian sub-basin and up to -5.7 in the Alboran sub-basin (Fig. 5d). The resolution of the available data hardly allows to evaluate the model performance for this water mass; nevertheless, station 74 (33°7N, 33°5E) in the Eastern Levantine basin exhibits a discernible radiogenic signal associated to LIW (more pronounced than the modelled one), while station 51 (33.5°N,

27°E) in the western Levantine basin does not reveal any specific isotopic signature. The surface waters originating from the Atlantic Ocean (Atlantic Waters, AW) are characterized by the most negative signature (value around -9) in good agreement with observations and are keeping the most negative signature over all the Med Sea, allowing a clear identification. The $\varepsilon_{Nd}$ signatures of the deep water mass display values around -6.5, consistent with the observations available in the Eastern basin (Fig. 5e).

Overall the model captures correctly the global vertical profiles of Nd isotopic signatures (averaged over the entire water column), especially producing a realistic and significant radiogenic signature associated to LIW at the Intermediate level(Fig. 6), although the $\varepsilon_{Nd}$ values are globally overestimated by roughly 1.5 $\varepsilon_{Nd}$ units.

## 4.4 The inter-annual variability

In this section, we analyse inter annual variations on the redistribution of $\varepsilon_{Nd}$ over the Mediterranean basin, with a special

focus on the possible impact of the EMT events. The evolution of the monthly averaged $\varepsilon_{Nd}$ at the intermediate level (between 200 and 600 m) in different "boxes" following the LIW trajectory from the Levantine sub-basin to the Algerian sub-basin (including Ionian, Sicily channel, Tyrrhenian, and Gulf of Lions) is represented in Fig. 7 for the 40 years of the simulation. It shows that $\varepsilon_{Nd}$ signatures vary seasonally with maximum amplitude of 0.2 $\varepsilon_{Nd}$ units. The EMT event significantly impacts the $\varepsilon_{Nd}$ signature at the global scale of the Med Sea. After 1992, which is referred as the beginning of the EMT event, an

important change of $\varepsilon_{Nd}$ distributions is simulated over all the Mediterranean Sea, with regional values shifted by almost 0.5 $\varepsilon_{Nd}$ units.

The drastic change caused by the EMT event at the beginning of 1990s is even more illustrated by showing the differences of $\varepsilon_{Nd}$ distributions between the pre-EMT situations in 1987 and the subsequent years up to 2010. The analysis on different



horizontal levels (Fig. 8) as well as along E-W sections (Fig. 9) provides a better understanding of the source of ventilation for the interior of the Mediterranean Sea, and the connection between the surface, intermediate and deep water redistribution.

In comparison with the steady state situation for the Mediterranean Sea circulation (pre-EMT), the surface waters are relatively less radiogenic in the Levantine sub-basin, the Algerian, and the Alboran sub-basins between 1995 and 1999 (Fig. 8a
and 8b). After 2001 these surface waters become more radiogenic over the whole basin. At the intermediate level only the EMed presents a less radiogenic signature in 1995, indeed the $\varepsilon_{Nd}$ are more radiogenic over the entire basin after 2001(Fig. 8f, 8g and 8h). The deep waters are globally more radiogenic between 1987 and 2010 especially in the EMed where increase of 1 units of $\varepsilon_{Nd}$ are simulated around the Aegean sub-basin. The vertical section illustrates the important penetration of the surface and intermediate waters characterized with radiogenic $\varepsilon_{Nd}$ into the deep waters near the Cretan Arc as a consequence
of the EMT that shifted the Nd isotopic signature by almost +1.3 $\varepsilon_{Nd}$ units in the bottom waters (Fig. 7b). This radiogenic signal is maximum in 1995 at the bottom water around the Cretan Arc near 26°E, and for the next years (i.e., 1997, 1999, 2005 and 2010) propagates in the deep waters of the whole Levantine sub-basin that becomes typically more radiogenic of + 0.5 of $\varepsilon_{Nd}$ (Fig. 9).

The amplifying tracer penetration caused by the EMT event generates less radiogenic values at the LIW layer in the EMed in
1995 and 1999, because this water is mixed with upwelled pre-EMT less radiogenic water masses. Contrastingly the simulated values become globally more radiogenic in the WMed. The radiogenic transient signal enters inside the western basin through the LIW outflow (up to +0.6 $\varepsilon_{Nd}$ unit) and gradually penetrates into the deep water through time. However the most $\varepsilon_{Nd}$ shift was simulates in the Levantine sub-basin deep water with more than 1.3 unit change (Fig. 7b and Fig. 9).

## 5 Discussion

The high resolution simulation presented here confirms that the main features of the $\varepsilon_{Nd}$ distribution are generated assuming that BE is the only Nd oceanic source term in the Med Sea, as previously demonstrated for the global ocean (Arsouze et al., 2007) and the Atlantic basin (Arsouze et al., 2010). This reinforces the preceding conclusions on BE as a major process in the Nd oceanic cycle, even at regional scale and in a semi-eclosed basin such as the Mediterranean basin. Although the processes leading to BE are still not fully understood yet (Jeandel and Oelkers, 2015), the resulting timescale is of the order of few
25  months, in agreement with Arsouze et al. (2010). This timescale is also consistent with the kinetic rates of Nd release from basaltic material during the batch experiments conducted by Pearce et al. (2013). It is also consistent with the field data and their Lagrangian modelling developed by Grenier et al. (2013) in the highly dynamic South West pacific. Taking into account the lithology of the margin sediments did not improve our simulations. This requires more laboratory experiments, targeted on the issue of the nature of the sediments. Nevertheless the comparison with the available data in the Med Sea reveals that this
approach simulates a slightly too radiogenic value in the surface and intermediate waters, especially in the EMed. This bias will likely be corrected once the dust and river inputs will be simulated. Indeed, those could locally affect the surface waters with less radiogenic values. The main river systems of the Mediterranean basin (i.e., the Nile, Po and Rhone) are characterized by a wide range of Nd IC signature, with an average $\varepsilon_{Nd}$ value of -10.2 for the Rhone, and rather radiogenic Nd isotopic ratios

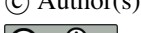



from the Nile ($\varepsilon_{Nd} \sim$ -4). The input of Saharan dust has important effects on the Mediterranean region (Guerzoni et al., 1997) where the Nd isotopic compositions of aerosols range from -9.2 in the eastern part of northern Africa (e.g., Egypt) to -16 in the central and western parts of northern Africa (Grousset and Biscaye, 2005; Scheuvens et al., 2013). Previous studies suggest that the $\varepsilon_{Nd}$ distribution at the near surface is for the most part reflecting river and aerosols inputs (Piepgras and Wasserburg,

1987; Arsouze et al., 2009; Jones et al., 2008; Siddall et al., 2008). Hence it is clear that taking into account the dust and river input in the future work could improve the simulation of Nd isotopic distribution in the Mediterranean Sea.

The LIW layer is particularly characterized by the most radiogenic signature in the intermediate level between 200 and 600 m, which is in good agreement with in-situ observations from Tachikawa et al. (2004) especially with the highest $\varepsilon_{Nd}$ value of -4.8 found at about 200 m in the easternmost Levantine basin.

The LIW represents the principal movement of water mass from the EMed into the WMed. The model simulates a satisfying lateral propagation into the western basin, demonstrated by a strong accumulation of radiogenic signal in the intermediate waters (Fig. 5d and 5e). This LIW signature is conserved in the WMed allowing us to study the impact of interannual variability, including the exceptional events observed in the ventilation of the deep waters (e.g., EMT) in the whole basin.

The sequence of the EMT events occurring in the EMed at the beginning of the 1990s has completely changed the deep

water mass structure. Different hypotheses concerning the preconditioning of the EMT and its timing have been proposed in the literature (Roether et al., 1996, 2007; Malanotte-Rizzoli et al., 1999; Lascaratos et al., 1999; Theocharis et al., 1992, 1999; Samuel et al., 1999; Theocharis and Kontoyiannis, 1999; Zervakis et al., 2000; Stanev and Peneva, 2002; Klein et al., 1999; Gertman et al., 2006; Josey, 2003). In our simulation, the $\varepsilon_{Nd}$ distribution between 1995 and 2001 also revealed that LIW and deep water signatures were very different from the pre-EMT picture (see Fig. 7). This amplification of mixing caused by

the EMT generates accumulation of radiogenic water at the bottom. The 1995 section emphasizes the severe impact of the EMT on water mass distribution, which transfers massive volumes of surface/intermediate waters into the deep layers, with the highest contributions toward the bottom and south of Crete (Fig. 8), causing a temporary change in the EMDW origin, from the Adriatic to Aegean sub-basin in 1992-1993. The renewal of the deep water masses is very similar to the tritium-helium3 redistribution observed by Roether et al. (2013) that is satisfyingly simulated by our regional model Ayache et al. (2015a). This

gives some more reliability to the evolution of $\varepsilon_{Nd}$ distribution simulated after the EMT event (Fig. 9).

The EMT modifies the characteristics of EMDW in the Levantine sub-basin by increasing the $\varepsilon_{Nd}$ signature over the entire Eastern basin (Fig. 9). Hence the LIW layer is also affected by this $\varepsilon_{Nd}$ shift, which is next transferred rapidly in the WMed by the overflow of the Sicily Channel. The LIW signal then propagates at depth in the western basin illustrating how the EMT event modify water masses characteristics and potentially affect the formation of deep and bottom water masses in this

sub-basin.

## 6   Conclusions

This study proposes a new map compiled from in-situ data with a sufficient resolution to cover the very complex morphology and geology of the Mediterranean Sea. This map shows Nd isotopic signatures for all the Mediterranean Sea margins. The



quality of this interpolated map allows using it as a continuous source of $\varepsilon_{Nd}$ to make a link between an ocean circulation model and the tracer inputs from the margins to better understand the thermohaline circulation in modern and paleo ocean circulation. This compilation provides a complete picture of the $\varepsilon_{Nd}$ of the whole Mediterranean margins which could interest other earth science fields (e.g., solid earth, weathering, tectonic, etc.).

The $\varepsilon_{Nd}$ distribution was simulated using a high-resolution regional model at 1/12° of horizontal resolution (6-8 km). The Boundary Exchange (BE) parameterization was performed via a relaxing term toward the isotopic composition of the margin from this new geological map. The characteristic margin-to-ocean exchange time is about 3 months in the Mediterranean Sea, in good agreement with the previous estimation of Arsouze et al. (2010) in the north Atlantic basin. Our next step was therefore to use a fully prognostic coupled dynamical/biogeochemical model with an explicit representation of all Nd sources (i.e.,

atmospheric dusts, dissolved river fluxes, and margin sediment re-dissolution) and sinks (i.e., scavenging) to simulate the Nd oceanic cycle in another dedicated study. More in-situ data (as those currently acquired in the framework of the GEOTRACES MEDBLACK programme) should help improving in the knowledge of Nd and its isotope cycles in the Med Sea to better constrained the fluxes of solid material and exchange between the continental margin and open ocean.

The Boundary Sources with $\varepsilon_{Nd}$ implemented as a passive conservative has represented an interesting opportunity to explore

the interannual variability on the $\varepsilon_{Nd}$ distribution. Indeed, the Eastern Mediterranean Transient (EMT) signal from the Aegean sub-basin was simulated, conducting to a significant and measurable evolution of eps Nd signal over the whole Mediterranean basin. It confirms that $\varepsilon_{Nd}$ represents an appropriate proxy to improve our knowledge on the long term trend in the Med Sea circulation, especially to explore if EMT-type events occurred in the past (Roether et al., 2014; Gačić et al., 2011). New Nd-paleo-data (e.g., Jiménez-Espejo et al., 2015) or recent Nd observations collected on corals or foraminifera in the context of

the PaleoMeX (Paleo Mediterranean Experiment) program should give the opportunity to address this question.

*Acknowledgements.* The authors wish to acknowledge F. Bassinot, Dr G. Tranchida and Dr P. Censi for sediments sample. We thank Dr S.Conticelli who kindly answered our questions on geological featuresof italian coast, which greatly helped for the interpolation. Hugo Pradalier is acknowledged for his contribution at the beginning of this work.



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





**Table 1.** Coordinates of the studied cores together with water depth .

| Cruise | Years | Longitude (°E) | Latitude (°N) | Depth (m) |
|---|---|---|---|---|
| ETNA80 | 1980 | 11.48 | 36.30 | 263 |
| | | 13,44 | 33,23 | 736 |
| DEDALE | 1987 | 25.59 | 33.51 | 3020 |
| NOE | 1984 | 30.01 | 32.19 | 1465 |
| | | 30.1 | 31.53 | 495 |
| Sicily strait | 2003 | 12.57 | 37.30 | 29.5 |
| | | 14.37 | 36.36 | 87.4 |
| | | 14.22 | 36.16 | 488.2 |
| | | 12.32 | 36.56 | 117 |

**Table 2.** Mean $\varepsilon_{Nd}$ for the Med Sea, from EXP3 and for the in-situ data from Tachikawa et al. (2004)

| | Model | | In-situ data | |
|---|---|---|---|---|
| | Intermediate waters | Average all depths | Intermediate waters | Average all depths |
| Mediterranean Sea | -5.8 | -6.2 | -7.6 ± 1.37 | -7.8 ± 1.54 |
| Eastern basin (EMed) | -4.7 | -5.1 | -7 ± 0.85 | -7.1 ± 1.08 |
| Western Basin (WMed) | -5.8 | -6.3 | -9.4 ± 0.69 | -9.6 ± 0.48 |

**Table 3.** Summary of the main characteristics for each experience

| Experience | Relaxing time $\tau$ | Regression coefficient for data/model points |
|---|---|---|
| EXP1 | 1 month | 0.37 |
| EXP2 | 2 months | 0.71 |
| EXP3 | 3 months | 0.61 |
| EXP4 | 1 year | 0.33 |
| EXP5 | $\tau$ varying vertically from 1 month at the surface to 10 months at 540 m | 0.29 |
| EXP6 | $\tau$ 1 month (max $\varepsilon Nd_{margin}$) to 1 year (min $\varepsilon Nd_{margin}$) | 0.40 |





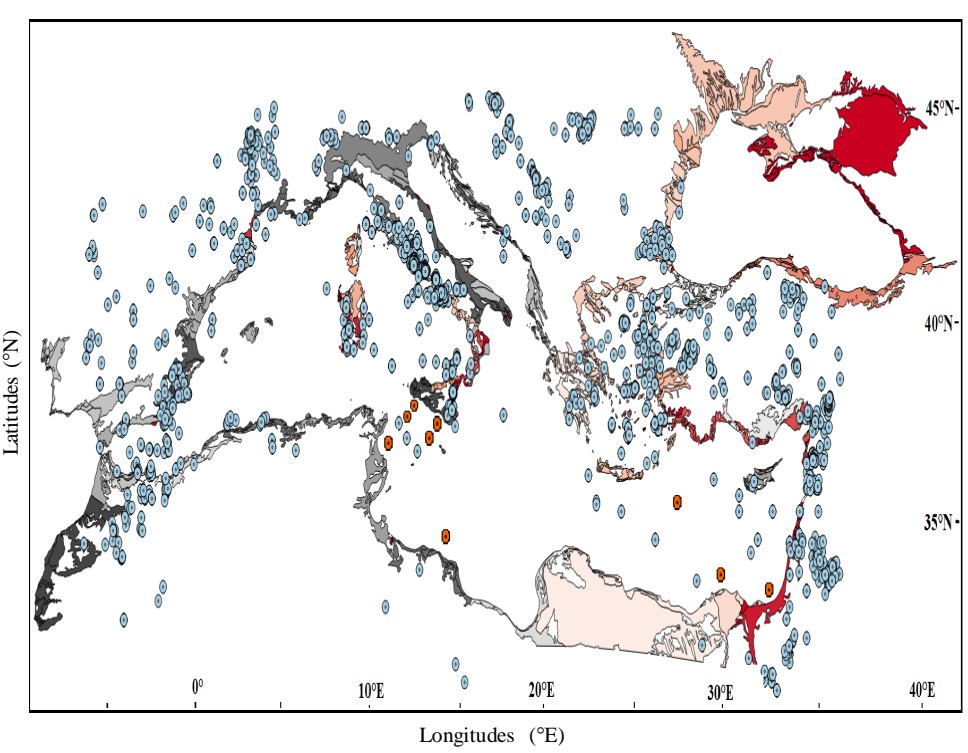

**Figure 1.** The filled contours indicate the geological province limit from a high resolution digital geological map (http://www.geologie.ens.fr/spiplabocnrs/spip.php?rubrique67while the circles filled in blue represent the location of the discrete data compiled from EarthChem database (see Appendix1), and in red the location of the stations corresponding to the sediments analysed as part of the present work.





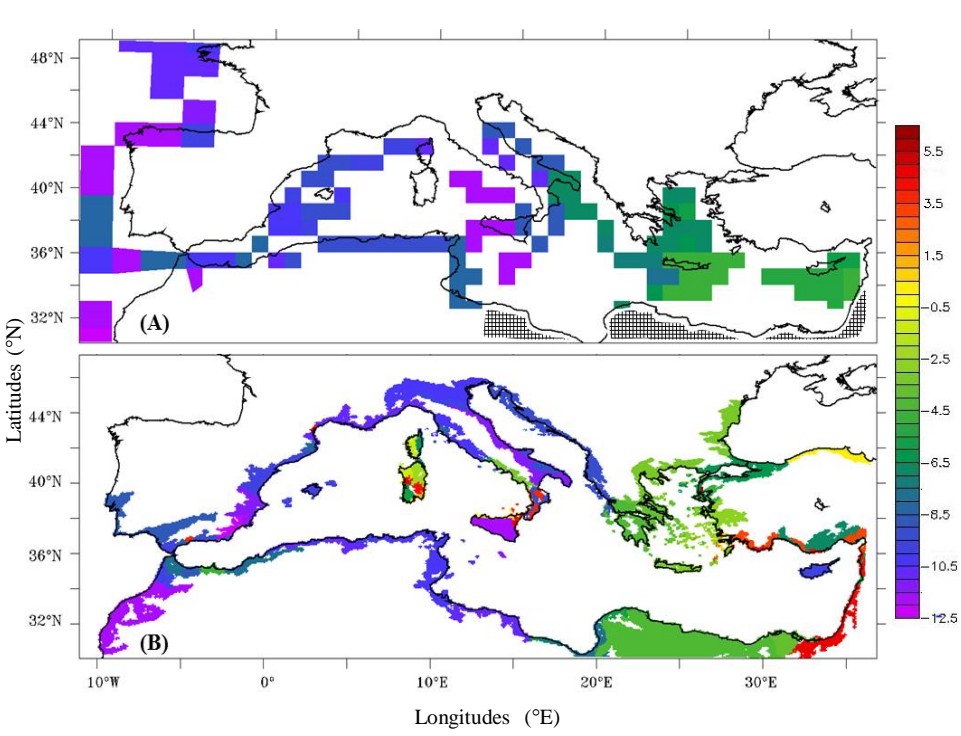

**Figure 2.** Extrapolated map providing a picture of the Nd signature of all the margins surrounding the Mediterranean Sea. (a) Low-resolution configuration ORCA2 (reproduced from Jeandel et al. (2007), and (b) high resolution configuration NEMO-MED12 (this work). Hatched areas correspond to uncharacterized areas in the published literature (before 2007) as done by Jeandel et al. (2007).





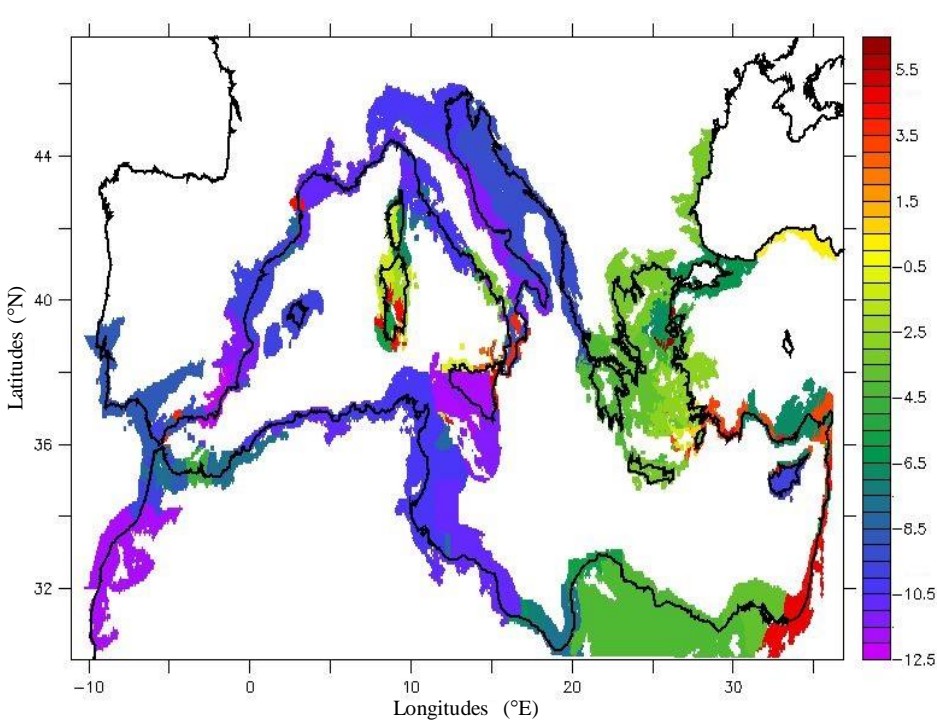

**Figure 3.** Map of the $\varepsilon Nd_{margin}$ used in the model simulation, done by interpolation of Fig. 1b on the oceanic margins of the Mediterranean Sea (see section 2).





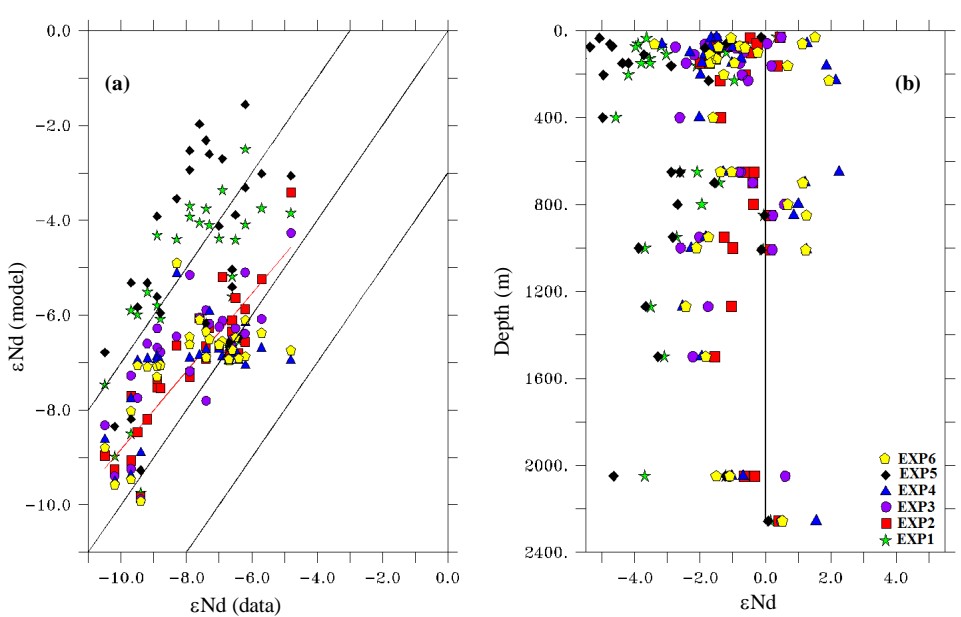

**Figure 4.** Model/data comparison for the 6 simulations performed with different relaxing time at the steady state (see Tab.3) and the in-situ data from Tachikawa et al. (2004): **(a)** model-data correlation, red line is the linear regression from EXP2. Diagonal black lines are lines $\varepsilon Nd$ (modeled) = $\varepsilon Nd$ (data), $\varepsilon Nd$ (modeled) = $\varepsilon Nd$ (data) + 3 $\varepsilon Nd$ and $\varepsilon Nd$ (modeled) = $\varepsilon Nd$ (data) -3 Tachikawa et al. (2004). **(b)** model/data comparison as a function of depth, dashed line represents the data from Tachikawa et al. (2004).





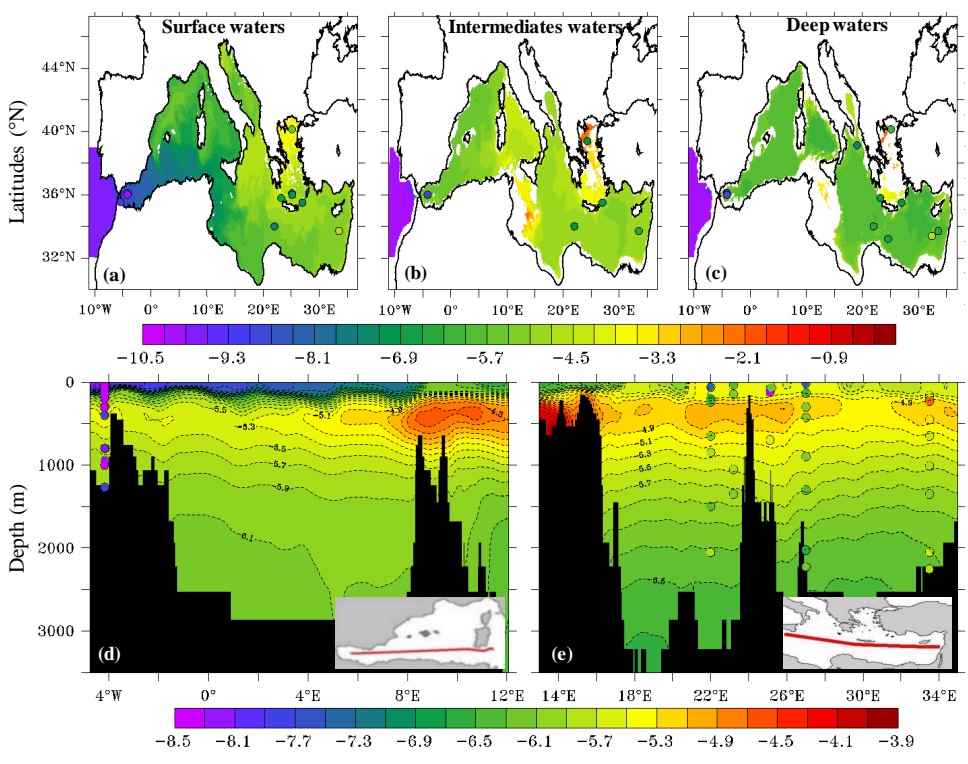

**Figure 5.** Output of model from EXP3 (t = 3 months) at the steady state. Upper panel: horizontal maps for surface waters **(a)**, intermediate waters **(b)**, and deep waters **(c)**. Lower panel E-W section in WMed **(d)**, and EMed **(e)**, whereas colour-filled dots represent in situ observations (Tachikawa et al., 2004). Both use the same colour scale.





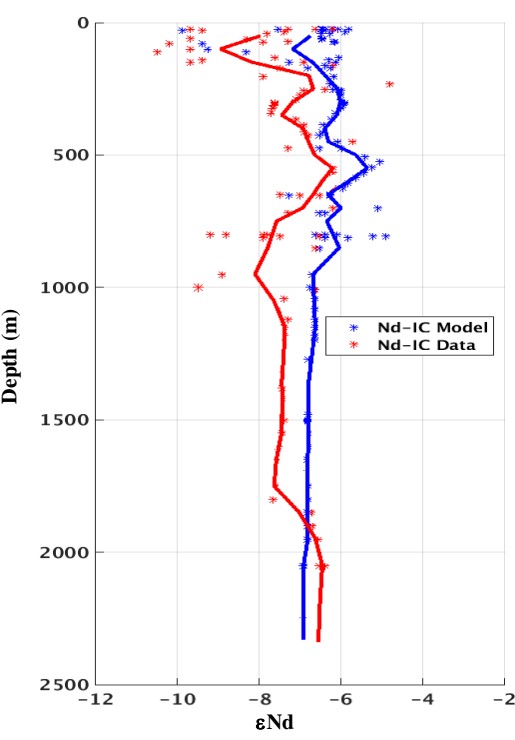

**Figure 6.** Comparison of average vertical profiles of Nd isotopic signature (Nd-IC) from EXP3 (t =3 months) in the whole Mediterranean Sea. Model results are in blue, while red indicates the in situ data.



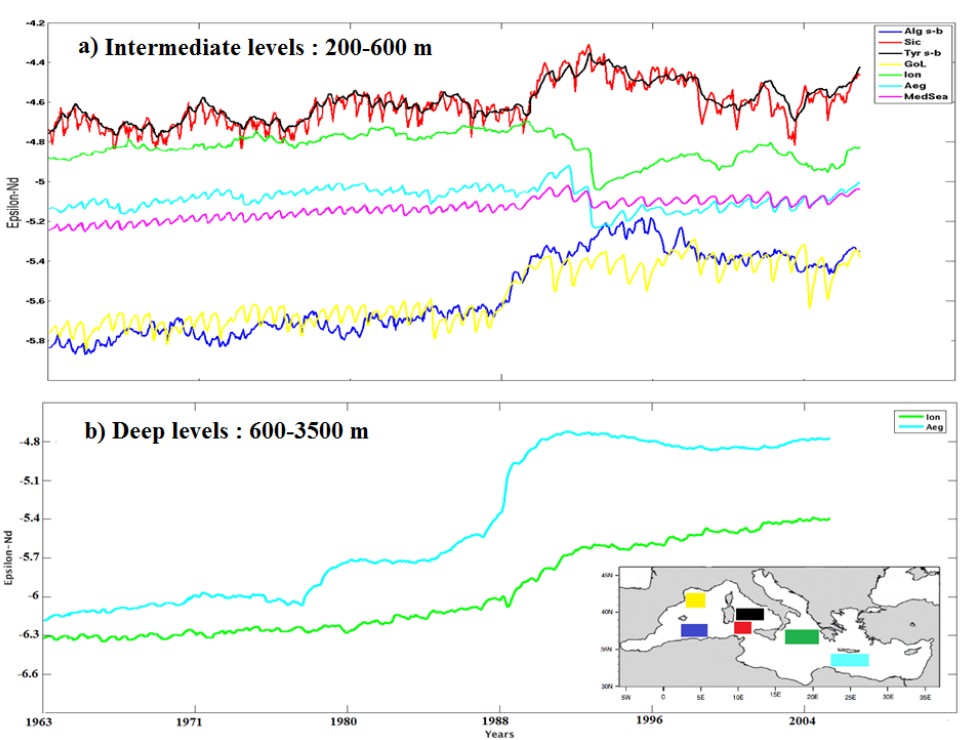

**Figure 7.** The $\varepsilon Nd$ evolution from EXP3 (tau = 3 months) at the intermediate level in **(a)** (average depth between 200 and 600 m), and for the deep levels in **(b)** (average depth between 600 and 3500 m). In the Gulf of Lions (yellow), Algerian sub-basin (blue), Levantine s-b (cyan), Ionian s-b (green), Sicily channel (red), Tyrrhenian s-b (black).





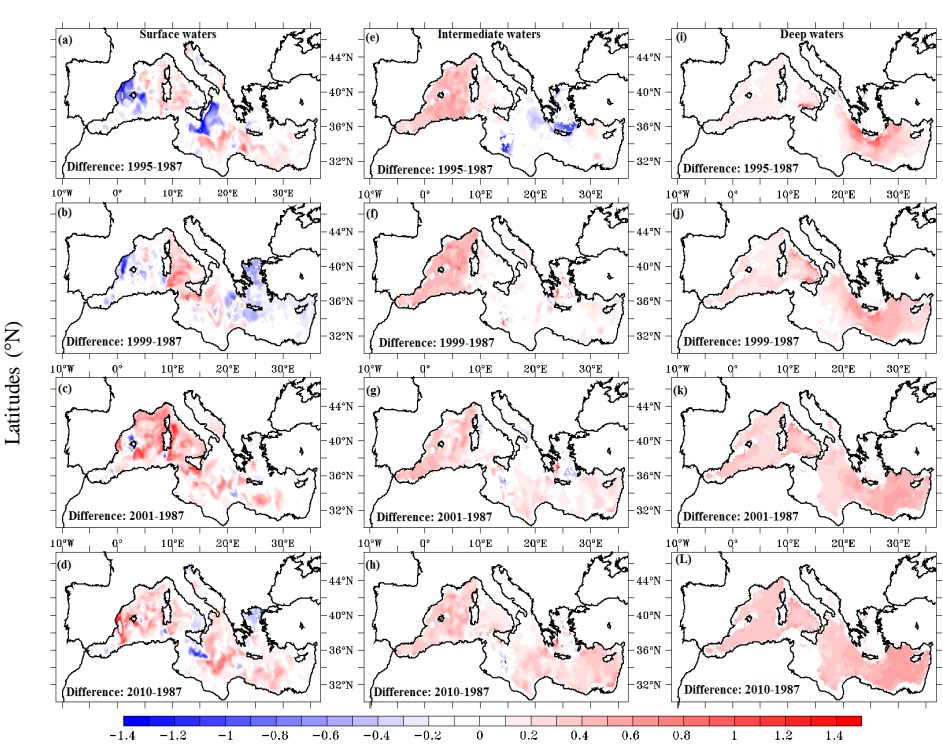

**Figure 8.** Horizontals maps showing the model output from EXP3 (tau = 3 month), in the left column for the surface level (0 - 200 m), in the middle column for the intermediate layer (between 250 - 600 m, and for the deep layer in the right column (600 - 3500 m).





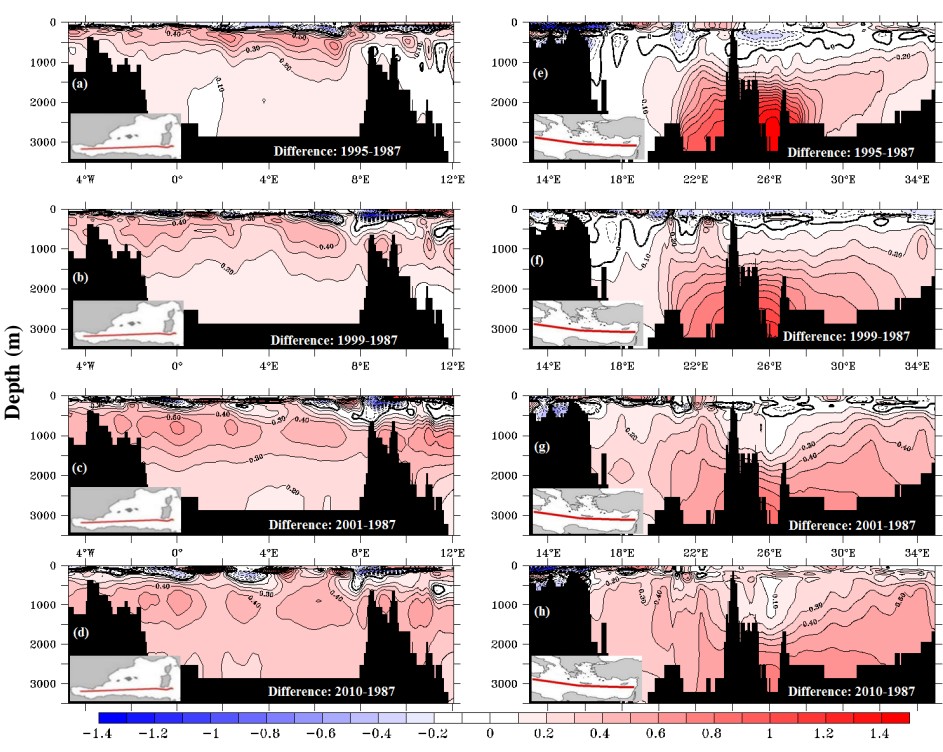

**Figure 9.** Colour filled contours represents simulated Nd isotopic composition for the WMed in the left column, and the EMed sections are shown in the right column. The first line show the situation in 1987 (pre-EMT), the others sections show the difference in the Nd-IC between 1995 and 2010 (post-EMT period) corrected to 1987 situation.