# Peer review of "High resolution neodymium characterization along the Mediterranean margins and modeling of $\varepsilon_{Nd}$ distribution in the Mediterranean basins."

_Biogeosciences, 2016_

## Referee Comment (RC1) · Anonymous Referee #1 · 1 May 2016

General comments This timely study presents (1) the high resolution map of Mediterranean margin eNd distribution, (2) the Mediterranean seawater eNd distribution simulated by the high-resolution regional model that was optimised for the relaxation term of boundary exchange (BE) and (3) the impact of Eastern Mediterranean Transient (EMT) on seawater eNd values simulated by the calibrated model. The compilation of continental margin Nd isotopic compositions is highly appreciable and extremely useful for studying present and past Mediterranean seawater eNd variability.

My major concern is the Nd modelling parts that used only BE as a source term. The

authors are aware that the simulated seawater eNd values are generally too radiogenic relative to measured seawater samples. They will integrate dust and river inputs as well as scavenging for their future modelling study. Under such condition, it is not clear the geochemical meaning of estimated Nd exchange time of three months. I would suggest that the authors add some more explanation for the following points.

1. Disagreement of LIW eNd values (eNd > -5) towards the western basin between the field data and the simulation. The advection of high eNd LIW (eNd > -5) towards the western basin has not been observed for the field data. I agree that seawater eNd data are still sparse to provide a robust diagnostic. But the authors ignored seawater Nd isotopic compositions reported by Henry et al. (1994) and Vance et al. (2004). The eNd value from site BAOR in the Strait of Sicily (eNd = -7.7±0.6; Henry et al., 1994) provides a constraint for eNd values of LIW entering the W Med Sea. Indeed, this value is consistent with other seawater Nd isotopic compositions from the intermediate water depths in the E Med Sea showing that high LIW values are confined in the easternmost part of the Levantine Sea (Fig. 5). Personally, I had never seen the LIW eNd values at sites close to the Strait of Sicily as high as -5 for the modern seawater and recent archives. I suggest that the authors consider all the existing Med seawater data to optimize the relaxation time (Figs. 4, 5 and 6) and add explanation about the possible reasons for the data-model decoupling for the LIW eNd values.

2. Diagnostic of simulation performance with vertical eNd profiles In relation to point 1, the comparison of vertical eNd profiles between the simulation and data would be carried out for site by site instead of the average profiles as shown in Figure 6. Seeing Figure 5d and 5e, one can have impression that simulated eNd is more stratified than the field data (for example Alboran Sea) but this feature does not appear in Figure 6. Since the number of sites providing seawater eNd profiles is limited, I suggest that the data-model comparison will be done for site by site along the longitude. This new Figure 6 will clarify the areas of data-model decoupling and will provide the elements of further discussion.

3. Too radiogenic eNd values for the Aegean Sea and its impact on the simulated EMT. The direct measurement of seawater samples indicates that Aegean seawater eNd values do not exceed -5.9±0.2 (Tachikawa et al., 2004). The simulated values are higher than this limit. The EMT simulation result could be sensitive to the performance Aegean Sea eNd simulation. Considering the overestimation of Aegean eNd values and too weak formation of the Adriatic Deep Water formation, the simulated shift of seawater Nd isotopic signatures due to EMT could be overestimated. It is true that more negative eNd values of surface water in southern part of the E Med Sea will be obtained by Saharan dust contribution. In contrast, it is not obvious that the Aegean Sea water eNd will decrease by considering Nd sources other than the BE. Since the simulated eNd shift due to the EMT is relatively small, some comments about this point would be added.

Taking into account the above-mentioned points, I am not totally convinced by the BE alone can explain the major features of the Mediterranean seawater eNd distribution. Nonetheless, this work provides an important advance of understanding seawater eNd distribution in the Mediterranean Sea. Considering the compilation effort and the originality of modelling aspects, I strongly recommend accepting this work after moderate revision.

Minor or specific comments Page 2, line 9, delete "all stable". Page 2, line 26. Define "IC". Page 2 footnote about the eNd definition. The equation should be corrected: eNd=[(143Nd/144Nd)sample/(143Nd/144Nd)CHUR-1]x10ˆ4 Page 3, line 5. The eNd value of the Mediterranean outflow was estimated to be -9.4 (Henry et al., 1994; Tachikawa et al., 2004). Page 5, line 16. Acid concentrations are shown with wrong fonts. "HN03" should be "HNO3". Page 6, line 9. Delete "Nd" after "eNd". Page 9, lines 9-10. "Lacan et al., (2012)" should be replaced by "Spivack and Wasserburg, 1988". Page 10, line 15. "underestimation" should be "overestimation". Page 11, line 5. About the comparison of intermediate eNd values between the model and the data. It is not clear the referred values here correspond to which part of Table 2. Page 14, line 16.

"eps Nd" should be corrected.

Figure 1. Indicate the relationship between the geological province and colour code in the caption. Figure 4. The x-axis should be delta eNd (data-model) and the y-axis should be "Water depth (m)" not "profondeur". There is no "dashed line" as indicated in the caption. It should be the black solid line. Figures 8 and 9. The colour code indicates eNd difference between specific year and 1987. Please indicate this information in the captions.

Appendix 2, 3 and 4. EXP5. The conditions of the relaxation time are different from Table 3. Please correct.

Reference Vance, D., Scrivner, A. E., Benecy, P., Staubwasser, M., and Henderson, G. M.: The use of foraminifera as a record of the past neodymium isotope composition of seawater, Paleoceanography, 19, doi:10.1029/2003PA000957, 2004.

---

## Referee Comment (RC2) · Anonymous Referee #2 · 8 Jun 2016

General comments

This MS deals with a modeling of $\varepsilon$Nd distribution in the Mediterranean based on the high resolution Nd concentration and IC databases along the Mediterranean margins. I greatly appreciate authors' tremendous effort to compile and establish database for model calculation. I admit that the model partly seems to reproduce the real distribution. I think, however, several issues should be clearly addressed before final publication.

Specific comments

**4.3 The $\varepsilon$Nd distribution**

Authors seems to insist that the main features of $\varepsilon$Nd distribution in the Mediterranean are well reproduced assuming only the BE operating Nd oceanic source. Authors also admit that the reproduced results are apparently more radiogenic than the in-situ data reported by Tachikawa et al. (2004). Although I partly agree that their model successfully reproduced some features of the distribution, I have two concerns on their approach. First, I am not quite sure whether the number and locality of in-situ data for comparison is sufficient or not. The data reported by Tachikawa do not cover whole the Mediterranean; the data are localized in the eastern and western parts, and almost no data in the central part. I do not think these data are sufficient for verifying the simulated results. I have found one depth profile at station Villefranche (43°24'N, 7°52'E) located in the central part (Henry et al., 1994), which seems to show great shifts from the simulated data. I do not understand why authors neglect this data and believe that authors should discuss the data. Second, in my opinion, the evaluation on contribution of dust and river inputs should be more quantitative. In discussion section (p12 L31), authors claim that an incorporation of dust and river inputs should solve the discrepancy between simulated results and in-situ data. This does not say anything because besides BE processes these two inputs exclusively control Nd flux. Although I agree that it is not so easy to incorporate dust and river inputs into simulation, authors is highly expected to add more quantitative comments on these inputs, say, how much additional Nd with low $\varepsilon$Nd is required to lower the simulated results.

**4.4 The inter-annual variability**

Although I found this is an interesting approach, I wonder how authors verify the results. Are there any marine samples recording the EMT events or any chances to observe the EMT in near future?

**Technical corrections**

p5 L16; "24" of "24N" should not be superscript. Also correct for HNO3 and HClO4.

p5 2.1.2; According to this section, the authors analyzed Nd IC of several sediment sample. Unfortunately, however, I could not understand how the Nd IC data are used in the model. I have checked Appendix 1 and could not find out. Please clarify this point.

p15-p17 References; Some unnecessary information, such as link to paper, is shown. Should be deleted.

p17 L10; This reference is from EPSL. Please write down the correct journal information.

Figure 5; "EXP3" should be "EXP2".

Appendix 1; What "$\lambda$" and "ÏŢ" stand for? Please explain. I could not find a list of reference for Appendix 1. Please add somewhere.

---

## Author Comment (AC1) · 12 Jun 2016

**High resolution neodymium characterization along the Mediterranean margins and modeling of εNd distribution in the Mediterranean basins.**

M. Ayache, J.-C. Dutay, T. Arsouze, J. Beuvier, S. Révillon, and C. Jeandel

Laboratoire des Sciences du Climat et de l'Environnement (LSCE), IPSL, CEA/UVSQ/CNRS, Orme des Merisiers, Gif-Sur-Yvette, France.

Correspondence to: M. Ayache (mohamed.ayache@lsce.ipsl.fr)

**We thank the anonymous reviewer #1 for her/his constructive comments on the manuscript. We have carefully considered all questions and concerns raised. The structure of our reply is as follows; each comment from the anonymous reviewer is recalled in blue, and our reply in black.**

1.Disagreement of LIW eNd values (eNd > -5) towards the western basin between the field data and the simulation. The advection of high eNd LIW (eNd > -5) towards the western basin has not been observed for the field data. I agree that seawater eNd data are still sparse to provide a robust diagnostic. But the authors ignored seawater Nd isotopic compositions reported by Henry et al. (1994) and Vance et al. (2004). The eNd value from site BAOR in the Strait of Sicily (eNd = -7.7 ±0.6; Henry et al., 1994) provides a constraint for eNd values of LIW entering the W Med Sea. Indeed, this value is consistent with other seawater Nd isotopic compositions from the intermediate water depths in the E Med Sea showing that high LIW values are confined in the easternmost part of the Levantine Sea (Fig. 5). Personally, I had never seen the LIW eNd values at sites close to the Strait of Sicily as high as -5 for the modern seawater and recent archives. I suggest that the authors consider all the existing Med seawater data to optimize the relaxation time (Figs. 4, 5 and 6) and add explanation about the possible reasons for the data-model decoupling for the LIW eNd values.

Our modeling approach gives a realistic simulation of eNd in the eastern part of Levantine basin (value up to -4.8, in agreement with in-situ data from Tachikawa et al., (2004)). However we agree with the referee that the simulated LIW IC are too radiogenic in the Ionian sub-basin and around the Sicily strait compared to in-situ observation, as mentioned in the paper ( see P1-L11; P11-L4; and P12-L30).

The purpose of this work is to test the impact of the BE on the Nd IC distribution in the Med Sea starting from the global expertise of Nd modeling by Arsouze et al., (2007, 2009) and by using a realistic representation of the margin Nd IC exclusively compiled from in situ data. Among all the sensitivity tests made on the calibration of this relaxing time, the EXP with 3 months produces the best agreement with available in-situ data.

We have used Tachikawa et al., (2004) and the recent unpublished data from the Meteor cruise M84 in 2011 (P. Montagnia et al, in prep, not showing here). All in-situ data for the

LIW layer gave an isotopic signature of almost -7 ± 1 (Tachikawa et al., 2004; Henry et al., 2004, and from P. Montagnia, in prep). Hence, we added the mentioned data from Henry et al. (1994) and Vance et al. (2004) to the statistic estimation of tau from Fig.4 (see the new Fig.4 and new Fig.5).

The model-data decoupling for the LIW layer can be explained by the fact that the LIW are formed in NW of Levantine sub-basin near the Cretan Arc, where the margin IC are about -4, leading to a relatively radiogenic signature  as we consider only the margin Nd source. Adding dust (eNd ~-12) and river inputs could likely improve the model performance for the representation of the LIW layer (on going work). Also, tritium/3He (Ayache et al., 2015) and CFC (Palmiéri et al., 2015) simulations have shown that the model overestimates the mixing near the Cretan Arc, and in consequence the Levantine sub-basin isotopic signature is over-represented in this water mass.

A sentence was added to the text in the revised manuscript about the possible reasons for the data-model decoupling for the LIW eNd values (see §5).

[Figure]

**Fig.4.** Model/data comparison for the 6 simulations performed with different relaxing time at the steady state (see Tab.3) and the in-situ data from Tachikawa et al., (2004) and Vance et al., (2004): **(a)** model-data correlation, red line is the linear regression from EXP2. Diagonal black lines are lines εNd (modeled) = εNd (data), εNd (modeled) = εNd (data) + 3 εNd and εNd (modeled) = εNd (data) −3 εNd. **(b)** model/data comparison as a function of depth, black solid line represents the data from Tachikawa et al., (2004) and Vance et al., (2004) and Henry et al., (1994).

[Figure]

**Fig.5.** Output of model from EXP3 (t =3 months) at the steady state. Upper panel: horizontal maps for surface waters (a), intermediate waters (b), and deep waters (c). Lower panel E-W section in WMed (d), and EMed (e), whereas colour-filled dots represent in situ observations (Tachikawa et al., 2004; Vance et al., 2004; Henry et al., 1994). Both use the same colour scale.

2. Diagnostic of simulation performance with vertical eNd profiles In relation to point 1, the comparison of vertical eNd profiles between the simulation and data would be carried out for site by site instead of the average profiles as shown in Figure 6. Seeing Figure 5d and 5e, one can have impression that simulated eNd is more stratified than the field data (for example Alboran Sea) but this feature does not appear in Figure 6. Since the number of sites providing seawater eNd profiles is limited, I suggest that the data-model comparison will be done for site by site along the longitude. This new Figure 6 will clarify the areas of data-model decoupling and will provide the elements of further discussion.

We agree with the referee that site by site data-model comparison gives a more interesting diagnostic of the simulation; however the available in-situ data are mainly localized in the Levantine sub-basin with one vertical profile in the Alboran sub-basin (see fig.5). In the revised manuscript we provide a new Figure 6 with a separation between these two sites (see Fig below).
The simulated values is realistic in the Alboran sub-basin surface water, but too radiogenic in the deep layer by more the 2.5 eNd units relative to in-situ data from Tachikawa et al. 2004. This shortcoming cold be associated with the Mediterranean outflow (mainly formed by the combination of LIW, EMDW and WMDW), where the Nd IC is overestimated in the LIW

layer (see response 1), and the southern penetration of new WMDW is weaker in the simulation compared to what was deduced from in situ observations, leads to excessively high eNd values at depth of Alboran sub-basin (Ayache., et al 2015).

Contrastingly the simulated isotopic signatures give a satisfying agreement with the data in the Levantine sub-basin leading globally to the same conclusion from the previous Fig.6 in the paper.

[Figure]

**Fig.6**: Comparison of average vertical profiles of Nd isotopic signature (Nd-IC) from EXP3 **(a)** in the Levantine sub-basin, **(b)** in the Alboran sub-basin, and **(c)** in the whole Mediterranean Sea. Model results are in blue, while red indicates the in situ data.

3. Too radiogenic eNd values for the Aegean Sea and its impact on the simulated EMT. The direct measurement of seawater samples indicates that Aegean seawater eNd values do not exceed -5.9 0.2 (Tachikawa et al., 2004). The simulated values are higher than this limit. The EMT simulation result could be sensitive to the performance Aegean Sea eNd simulation. Considering the overestimation of Aegean eNd values and too weak formation of the Adriatic Deep Water formation, the simulated shift of seawater Nd isotopic signatures due to EMT could be overestimated. It is true that more negative eNd values of surface water in southern part of the E Med Sea will be obtained by Saharan dust contribution. In contrast, it is not obvious that the Aegean Sea water eNd will decrease by considering Nd sources other than the BE. Since the simulated eNd shift due to the EMT is relatively small, some comments about this point would be added.

As explained in response to the first comment, we agree that the model simulates too radiogenic values compared to in-situ data. Nevertheless, our calculations from this sensitivity test on the EMT are not affected by the "initial" value in the Aegean sub-basin because our purpose here is to evaluate the impact of a like-EMT event on the Nd IC in the EMed. Hence our results suggest that the shift is more important in the Levantine deep water, compared to intermediate water where the EMT impact is lower.

One of our goals is to give a useful diagnostic on the long term variability of Med Sea circulation and to demonstrate the potential of Nd to detect a like-EMT event. However we agree with the referee that the weak formation of AdDW could affect the simulated sift of seawater Nd IC.

For the sake of clarity specific paragraph has been added to the text (see §5)

Taking into account the above-mentioned points, I am not totally convinced by the BE alone can explain the major features of the Mediterranean seawater eNd distribution. Nonetheless, this work provides an important advance of understanding seawater eNd distribution in the Mediterranean Sea. Considering the compilation effort and the originality of modelling aspects, I strongly recommend accepting this work after moderate revision.

Many thanks for these positive remarks.
.....

Minor or specific comments

Page 2, line 9, delete "all stable".
Done

Page 2, line 26. Define "IC".
Done Nd IC => Nd Isotopic Composition

Page 2 footnote about the eNd definition.

The equation should be corrected: eNd=[(143Nd/144Nd)sample/(143Nd/144Nd)CHUR-1]x10^4
Corrected

Page 3, line 5. The eNd value of the Mediterranean outflow was estimated to be -9.4 (Henry et al., 1994; Tachikawa et al., 2004).
Changed

Page 5, line 16. Acid concentrations are shown with wrong fonts. "HN03" should be "HNO3".
Corrected

Page 6, line 9. Delete "Nd" after "eNd".
Done

Page 9, lines 9-10. "Lacan et al., (2012)" should be replaced by "Spivack and Wasserburg, 1988".
Replace, we thank the referee for this suggestion

Page 10, line 15. "underestimation" should be "overestimation".
Done

Page 11, line 5. About the comparison of intermediate eNd values between the model and the data. It is not clear the referred values here correspond to which part of Table 2.

Corrected to: However averaged simulated values are relatively too radiogenic at the intermediate level (-5.8 compared to **-9.4± 0.69**, Tab.2).

Page 14, line 16."eps Nd" should be corrected.
Done

Figure 1. Indicate the relationship between the geological province and colour code in the caption.

Done, see change below:

The filled contours indicate the geological province limit based on the geological age (i.e. each color represent a separate age) from a high resolution digital geological map (http://www.geologie.ens.fr/spiplabocnrs/spip.php?rubrique67while the circles filled in blue represent the location of the discrete data compiled from EarthChem database (see Appendix1), and in red the location of the stations corresponding to the sediments analysed as part of the present work

Figure 4. The x-axis should be delta eNd (data-model) and the y-axis should be "Water depth (m)" not "profondeur". There is no "dashed line" as indicated in the caption. It should be the black solid line.
Done.

Figures 8 and 9. The colour code indicates eNd difference between specific year and 1987. Please indicate this information in the captions.
Done.

Appendix 2, 3 and 4. EXP5. The conditions of the relaxation time are different from Table 3. Please correct.
Corrected

We would like to thank you for the mentioned references; we will introduce the references in the introduction section.

**References**

Arsouze, T., Dutay, J. C., Lacan, F., and Jeandel, C.: Modeling the neodymium isotopic composition with a global ocean circulation model,Chemical Geology, 239, 165–177, doi:10.1016/j.chemgeo.2006.12.006, 2007

Arsouze, T., Dutay, J.-C., Lacan, F., and Jeandel, C.: Reconstructing the Nd oceanic cycle using a coupled dynamical – biogeochemical model, doi:10.5194/bgd-6-5549-2009, 2009

Ayache, M., Dutay, J.-C., Jean-Baptiste, P., Beranger, K., Arsouze, T., Beuvier, J., Palmieri, J., Le-vu, B., and Roether, W.: Modelling of the anthropogenic tritium transient and its decay product helium-3 in the Mediterranean Sea using a high-resolution regional model, Ocean Science, 11, 323–342, doi:10.5194/os-11-323-2015,http://www.oceansci.net/11/323/2015/os-11-323-2015.html, 2015

Henry, F., Jeandel, C., Dupré, B., and Minster, J.-F.: Particulate and dissolved Nd in the western Mediterranean Sea: Sources, fate and budget, Marine Chemistry, 45, 283–305, doi:10.1016/0304-4203(94)90075-2, 1994

Tachikawa, K., Roy-Barman, M., Michard, A., Thouron, D., Yeghicheyan, D., and Jeandel, C.: Neodymium isotopes in the Mediterranean Sea: Comparison between seawater and sediment signals, Geochimica et Cosmochimica Acta, 68, 3095–3106,doi:10.1016/j.gca.2004.01.024, 2004

Vance, D., Scrivner, A. E., Benecy, P., Staubwasser, M., and Henderson, G. M.: The use of foraminifera as a record of the past neodymium isotope composition of seawater, Paleoceanography, 19, doi:10.1029/2003PA000957, 2004.

---

## Author Comment (AC2) · 12 Jun 2016

**High resolution neodymium characterization along the Mediterranean margins and modeling of εNd distribution in the Mediterranean basins.**

M. Ayache, J.-C. Dutay, T. Arsouze, J. Beuvier, S. Révillon, and C. Jeandel

Laboratoire des Sciences du Climat et de l'Environnement (LSCE), IPSL, CEA/UVSQ/CNRS, Orme des Merisiers, Gif-Sur-Yvette, France.

Correspondence to: M. Ayache (mohamed.ayache@lsce.ipsl.fr)

**We thank the anonymous reviewer #2 for her/his constructive comments on the manuscript. We have carefully considered all questions and concerns raised. The structure of our reply is as follows; each comment from the anonymous reviewer is recalled in blue, and our reply in black.**

**4.3 The εNd distribution**

Authors seems to insist that the main features of εNd distribution in the Mediterranean are well reproduced assuming only the BE operating Nd oceanic source. Authors also admit that the reproduced results are apparently more radiogenic than the in-situ data reported by Tachikawa et al. (2004). Although I partly agree that their model successfully reproduced some features of the distribution, I have two concerns on their approach. First, I am not quite sure whether the number and locality of in-situ data for comparison is sufficient or not. The data reported by Tachikawa do not cover whole the Mediterranean; the data are localized in the eastern and western parts, and almost no data in the central part. I do not think these data are sufficient for verifying the simulated results. I have found one depth profile at station Ville-franche (43° 24'N,7° 52'E) located in the central part (Henry et al., 1994), which seems to show great shifts from the simulated data. I do not understand why authors neglect this data and believe that authors should discuss the data. Second, in my opinion, the evaluation on contribution of dust and river inputs should be more quantitative. In discussion section (p12 L31), authors claim that an incorporation of dust and river inputs should solve the discrepancy between simulated results and in-situ data. This does not say anything because besides BE processes these two inputs exclusively control Nd flux. Although I agree that it is not so easy to incorporate dust and river inputs into simulation, authors is highly expected to add more quantitative comments on these inputs, say, how much additional Nd with low εNd is required to lower the simulated results.

We agree with the referee that more in-situ data (as those currently acquired in the framework of the GEOTRACES MEDBLACK programme) should help improving in the knowledge of Nd and its isotope cycles in the Med Sea to better constrained the fluxes of solid material and exchange between the continental margin and open ocean, as mentioned in the paper (see P14-L11-14). In this study we have evaluated the model result against published observations: Tachikawa et al., 2004 d, , and following the reviewer suggestion we added the mentioned data from Henry et al. (1994) and Vance et al. (2004) to the statistic estimation of

tau from Fig.4 (see the new Fig.4 and new Fig.5). We added the station of Ville-franche (43° 24'N,7° 52'E) from Henry et al. (1994) in Fig.5a, 5b et 5d (see the new Fig.5). However this vertical profile is not showed in Fig.5d, 5e, because it is situated far away from the E-W section in the southern part of western basin. Some new data from Meteor 84/3 cruise in 2011 (Montagna et al,. in prep) will soon be available, but as they are still not published we could not use them in our manuscript. However, confronting these new data with our model experiment do not change any of our conclusions (personal communication).

Second, the purpose of this work is to test the impact of the BE on the Nd IC distribution in the Med Sea starting from the global expertise of Nd modeling by Arsouze et al., (2007, 2009) and by using a realistic representation of the margin Nd IC exclusively compiled from in situ data. Nevertheless this approach simulates a too radiogenic value in the Med Sea; this bias will likely be corrected once the dust and river inputs will be included in the model. As the reviewer mentioned, is not so easy to incorporate dust and river inputs into simulation, because this requires another modeling approach completely different from the adopted method here (i.e. we should simulate the Nd total concentration instead of εNd, and a fully prognostic coupled dynamical/biogeochemical model to represent the scavenging of Nd in the surface water and the remineralisation in the deep layer). This is an ongoing work which aims at explicitly representing and quantifying the different sources and sinks implied in the oceanic cycle of the Nd. The results of this second approach will be addressed in a coming paper.

A sentence was added to the text in the revised manuscript about the possible reasons for the data-model decoupling for the LIW eNd values (see §5).

[Figure]

**Fig.4.** Model/data comparison for the 6 simulations performed with different relaxing time at the steady state (see Tab.3) and the in-situ data from Tachikawa et al., (2004) and Vance et

al., (2004): **(a)** model-data correlation, red line is the linear regression from EXP2. Diagonal black lines are lines εNd (modeled) = εNd (data), εNd (modeled) = εNd (data) + 3 εNd and εNd (modeled) = εNd (data) −3 εNd. **(b)** model/data comparison as a function of depth, black solid line represents the data from Tachikawa et al., (2004) and Vance et al., (2004) and Henry et al., (1994).

[Figure]

**Fig.5.** Output of model from EXP3 (t =3 months) at the steady state. Upper panel: horizontal maps for surface waters (a), intermediate waters (b), and deep waters (c). Lower panel E-W section in WMed (d), and EMed (e), whereas colour-filled dots represent in situ observations (Tachikawa et al., 2004; Vance et al., 2004; Henry et al., 1994). Both use the same colour scale.

**4.4 The inter-annual variability**
Although I found this is an interesting approach, I wonder how authors verify the results. Are there any marine samples recording the EMT events or any chances to observe the EMT in near future?

As detailed by Roether et al. (1996, 2007), the EMT was a temporary change in the EMDW formation that occurred when the source of this deep water switched from the Adriatic Sea to the Aegean Sea during 1992–1993. In a previous evaluation of the same dynamical model used in this study (Ayache et al., 2015), we have shown that the NEMO-MED12 model

simulate correctly the EMT even with its corresponding penetration of tracers into the deep water in early 1995. Especially the transient evolution of the tracer age in the eastern basin revealed that the renewal of the bottom water masses is correctly simulated after the EMT (Ayache et al., 2015).

Hence, and by using the same model circulation (as in Ayache et al., 2015), and starting from the simulated steady state distribution of εNd, we have made this sensitivity test to see the response of Nd IC to this well documented event of EMT, as showed in Fig. 7 and Fig.8. This test gives a useful diagnostic on the long term variability of Med Sea to explore if an EMT-type event occurred in the past (Roether et al., 2014; Gacic et al., 2011).

Nd measurements will be performed on deep and intermediate corals in the med sea (Paleomex project, Mistrals/France). These new data will provide some information about past Mediterranean circulation, for more than hundred years ago. Our modeling efforts will represent a support to detect if some "EMT-like" events have occurred in the past.

Technical corrections

p5 L16; "24" of "24N" should not be superscript. Also correct for HNO3 and HClO4

Done (see § 2.1.2).

p5 2.1.2; According to this section, the authors analyzed Nd IC of several sediment sample. Unfortunately, however, I could not understand how the Nd IC data are used in the model. I have checked Appendix 1 and could not find out. Please clarify this point.

Additional analyses of surface sediments were done, in order to improve the EarthChem dataset in key areas (e.g., Sicilian strait). And to complete this dataset in the areas with low spatial resolution of available data, the localization of this stations are showed in red in Fig.1.

For the sake of clarity a new column has been added to table. 1, with the results of the analyzed sediment samples (see new Tab.1), this core tops of sediments collected along a given margin (Fig.1) are directly in contact with the water masses and therefore representative of the signatures of the sediments supposed to "contaminate" the waters flowing along them. We applied the resulted isotopic signature analyzed in this study, in the Sicilo-Tunisian, Libyan, and Egyptian margins age based on the Nd model-age relationships (Allegre,2005; Goldstein et al., 1984, 1997; O'Nions et al., 1979).

**Tab.1.** Coordinates and results of the studied cores together with water depth

| Cruise | Years | Longitude (°E) | Latitude (°N) | Depth (m) | $\varepsilon_{Nd}$ |
|--------|-------|----------------|---------------|-----------|---------|
| ETNA80 | 1980 | 11.48 | 36.30 | 263 | -10.09 |
|  |  | 13.44 | 33,23 | 736 | -10.92 |
| DEDALE | 1987 | 25.59 | 33.51 | 3020 | -8.15 |
| NOE | 1984 | 30.01 | 32.19 | 1465 | -4.49 |
|  |  | 30.1 | 31.53 | 495 | -3.92 |
| Sicily strait | 2003 | 12.57 | 37.30 | 29.5 | -11.67 |
|  |  | 14.37 | 36.36 | 87.4 | -11.05 |
|  |  | 14.22 | 36.16 | 488.2 | -11.22 |
|  |  | 12.32 | 36.56 | 117 | -8.06 |

p15-p17 References; Some unnecessary information, such as link to paper, is shown. Should be deleted.
Done (see references)

p17 L10; This reference is from EPSL. Please write down the correct journal information.
 Herrmann, M. J. and Somot, S, 2008 paper is published in Geophysical Research letters

Figure 5; "EXP3" should be "EXP2".

Corrected

Appendix 1; What "" and "Ï¸T" stand for? Please explain. I could not find a list of reference for Appendix 1. Please add somewhere.

We have added the list of reference used in Appendix 1, in a new Excel file (more than 600 references)

λ for longitude and φ for latitudes, corrected in the new version.

**References**

Allegre, C.: Géologie Isotopique., Belin ed, Paris, 2005.

Arsouze, T., Dutay, J. C., Lacan, F., and Jeandel, C.: Modeling the neodymium isotopic composition with a global ocean circulation model,Chemical Geology, 239, 165–177, doi:10.1016/j.chemgeo.2006.12.006, 2007

Arsouze, T., Dutay, J.-C., Lacan, F., and Jeandel, C.: Reconstructing the Nd oceanic cycle using a coupled dynamical – biogeochemical model, doi:10.5194/bgd-6-5549-2009, 2009

Ayache, M., Dutay, J.-C., Jean-Baptiste, P., Beranger, K., Arsouze, T., Beuvier, J., Palmieri, J., Le-vu, B., and Roether, W.: Modelling of the anthropogenic tritium transient and its decay product helium-3 in the Mediterranean Sea using a high-resolution regional model, Ocean Science, 11, 323–342, doi:10.5194/os-11-323-2015, 2015

Henry, F., Jeandel, C., Dupré, B., and Minster, J.-F.: Particulate and dissolved Nd in the western Mediterranean Sea: Sources, fate and budget, Marine Chemistry, 45, 283–305, doi:10.1016/0304-4203(94)90075-2, 1994

Gacic, M., Civitarese, G., Eusebi Borzelli, G. L., Kova cevic, V., Poulain, P.-M., Theocharis, A., Menna, M., Catucci, A., and Zarokanellos, N.: On the relationship between the decadal oscillations of the northern Ionian Sea and the salinity distributions in the eastern Mediterranean, Journal of Geophysical Research, 116, C12 002, doi:10.1029/2011JC007280, 2011.

Goldstein, S., O'Nions, R., and Hamilton, P.: A Sm-Nd isotopic study of atmospheric dusts and particulates from major river systems, Earth and Planetary Science Letters, 70, 221–236, doi:10.1016/0012-821X(84)90007-4, 1984.

Goldstein, S., Arndt, N., and Stallard, R.: The history of a continent from U-Pb ages of zircons from Orinoco River sand and Sm-Nd isotopes in Orinoco basin river sediments, Chemical Geology, 139, 271–286, doi:10.1016/S0009-2541(97)00039-9, 1997.

O'Nions, R. K., Evensen, N. M., and Hamilton, P. J.: Geochemical modeling of mantle differentiation and crustal growth, Journal of Geophysical Research, 84, 6091, doi:10.1029/JB084iB11p0609, 1979.

Roether, W., Manca, B. B., Klein, B., Bregant, D., Georgopoulos, D., Beitzel, V., Kovacevic, V., and Luchetta, A.: Recent Changes in Eastern Mediterranean Deep Waters, Science, 271, 333–335, doi:10.1126/science.271.5247.333, 1996

Roether, W., Klein, B., Manca, B. B., Theocharis, A., and Kioroglou, S.: Transient Eastern Mediterranean deep waters in response to the massive dense-water output of the Aegean Sea in the 1990s, Progress in Oceanography, 74, 540–571, doi:10.1016/j.pocean.2007, 2007

Tachikawa, K., Roy-Barman, M., Michard, A., Thouron, D., Yeghicheyan, D., and Jeandel, C.: Neodymium isotopes in the Mediterranean Sea: Comparison between seawater and sediment signals, Geochimica et Cosmochimica Acta, 68, 3095–3106, doi:10.1016/j.gca.2004.01.024, 2004

Vance, D., Scrivner, A. E., Benecy, P., Staubwasser, M., and Henderson, G. M.: The use of foraminifera as a record of the past neodymium isotope composition of seawater, Paleoceanography, 19, doi:10.1029/2003PA000957, 2004.

---

## Editor Decision (ED1)

Referee #1

I appreciate the new Fig. 6 that shows detailed eNd comparison between model and data for different basins. Considering this new observation, I would suggest that the authors revise the text corresponding to the performance of simulation. The data-model comparison rather indicate that the BE is not sufficient to reproduce the modern Mediterranean seawater eNd distribution. The decoupling is significant, not only for LIW but also in the Alboran Sea. I agreed with the authors' answer to my comments ("the LIW layer gave an isotopic signature of almost -7 ± 1; Tachikawa et al., 2004; Henry et al., 2004, and from P. Montagnia, in prep.") but the revised version is not always consistent with their answer.

I strongly recommend that the following parts would be modified before the final acceptance of this work.

P. 4, line 8 and throughout the text, as well as the figure caption. The compilation by Tachikawa et al. (2004) does not include Vance et al. (2004). This reference should be cited systematically for the data-model eNd comparison.
P. 4, line 21, "extrapolate". For the eNd compilation of margins, I expect that both extrapolation and interpolation were applied.
P.10, line 13, about correlation coefficient shown in Table 3. The coefficients shown in the text (0.71 and 0.61) do not correspond to the values indicated in Table 3. Please check and correct them.
P. 10, line 14, "dashed line" that is not shown in Fig. 4. I had already mentioned this point in my previous review. Please correct it.
P. 10, lines 15-16, "a slight overestimation of eNd between 0.3 and 1 eNd unites". According to the new Fig. 4b, the size of the offset should be larger (ex. the offset seems to be -2 at around 200m). Please check.
P. 10, line 18, "reasonable East-West gradient of eNd". What does "reasonable" mean? It is necessary to change this ambiguous expression.
P.11, lines 5-6. The eNd overestimation at the intermediate water depths is not limited for the Sicily Strait and Tyrrhenian sub-basins. The Alboran Sea presents a large offset (Fig. 6). Also "but the lack of observations prevent us to assess their consistency" is not appropriate and inconsistent with the authors answer ("the LIW layer gave an isotopic signature of almost -7 ± 1; Tachikawa et al., 2004; Henry et al., 2004, and from P. Montagnia, in prep.").
P.11, line 15. Please clarify "any specific isotopic signature".
P. 11, lines 20-22, "Overall the model capture correctly…". This part should be revised taking into account the observed offset between the simulated and measured seawater eNd distribution.
P. 12, lines 20-21, "The high resolution… in the Med Sea". This part should be nuanced.
P. 12, line 30, "especially in the EMed". The statement is not totally correct because of the strong offset in the Alboran Sea.
P. 13, lines 7-13. "The LIW layer is …in the whole basin". This part should be modified as I suggest at the beginning of this review.

Figure 4. Label "a" and "b" is missing. On the x-axis of Fig. 4b, "epsilon" is not correctly shown.
Figure 5. I am not sure that data from Henry et al. (1994) and Vance et al. (2004) are

shown. Please check and correct, if necessary.
Figure 6. I would put the Alboran result on the left, the Levantine result in the middle because of their east-west position in the Mediterranean Sea. "Tachikawa et al., 1983" should be "Tachikawa et al., 2004". Add also the other references (ex. Vance et al., 2004; Henry et al., 1994) when necessary.
Figure 7a and 7b. The cyan curves show Levantine eNd variation, not "Aeg".

Reference
Vance, D., Scrivner, A. E., Benecy, P., Staubwasser, M., and Henderson, G. M.: The use of foraminifera as a record of the past neodymium isotope composition of seawater, Paleoceanography, 19, doi:10.1029/2003PA000957, 2004.
* * *
Referee #2

General comments
This is a revised manuscript of Ayache et al. Along with the comments give by the referees, the authors seem to significantly improve the manuscript. I think, however, a couple of issues are still remained to be addressed before final publication.

Specific comments
After I read this manuscript, the following idea came to my mind: I wonder what is the aim of this study?
As written in the first sentence of discussion section, one important finding of this study is that the main features of the Nd IC distribution in the Mediterranean Sea are generated by assuming BE as the only Nd oceanic source term, which has been already demonstrated for the global ocean and the Atlantic basin. This fact confuses me a lot, because the authors have already found that their approach could be applied to much wider oceanic regions than the Mediterranean Sea. Furthermore, the Mediterranean Sea is a semi-closed basin and seems to be much easier system for modeling study than the global ocean. Therefore, I am not quite sure why they studied this oceanic region at this moment, which should have been done much earlier stage. I admit that more detailed and precise geological information might be available around the Mediterranean Sea than the global ocean, and this would lead to facilitate a more accurate simulation on Nd IC distribution. In reality, however, according to this manuscript, some problems (more radiogenic values in some areas etc.) still remain to be solved for a realistic simulation. Therefore, I recommend the authors to emphasize what is the merit to do modeling work for the Mediterranean Sea comparing with the global ocean. Otherwise, this paper would only deal with a case study of a limited oceanic region.

Technical corrections

P4 L28; What is "rive"? Please correct.

P7 L28; "(Ludwig et al., 2009)" should be "Ludwig et al. (2009)".

P14 L15; "Our next step is" seems to be much proper than "Our next step was".

Figs 4 and 5; Although the authors show new figures in the documents for replying referees' comments, those figures presented in the revised manuscript are the previous ones. Please replace to the new ones.

---

## Author Response (AR2)

**High resolution neodymium characterization along the Mediterranean margins and modeling of εNd distribution in the Mediterranean basins.**

M. Ayache, J.-C. Dutay, T. Arsouze, J. Beuvier, S. Révillon, and C. Jeandel

Laboratoire des Sciences du Climat et de l'Environnement (LSCE), IPSL, CEA/UVSQ/CNRS, Orme des Merisiers, Gif-Sur-Yvette, France.

Correspondence to: M. Ayache (mohamed.ayache@lsce.ipsl.fr)

We thank Koji Suzuki (Editor), and anonymous Referees for their constructive comments and suggestions, which have helped to improve the manuscript. We have carefully considered all questions and concerns raised.

The structure of our reply is as follows; each comment from the anonymous reviewer is recalled in blue, and our reply in black.

We provide a marked-up manuscript version; all change is recalled in red (see the revised manuscript with point-by-point responses below, after the responses to the reviewers' comments).

**Reply to Referree#1**

I appreciate the new Fig. 6 that shows detailed eNd comparison between model and data for different basins. Considering this new observation, I would suggest that the authors revise the text corresponding to the performance of simulation. The data model comparison rather indicate that the BE is not sufficient to reproduce the modern Mediterranean seawater eNd distribution. The decoupling is significant, not only for LIW but also in the Alboran Sea. I agreed with the authors' answer to my comments ("the LIW layer gave an isotopic signature of almost -7 ± 1; Tachikawa et al., 2004; Henry et al., 2004, and from P. Montagnia, in prep.") but the revised version is not always consistent with their answer.
I strongly recommend that the following parts would be modified before the final acceptance of this work.
We agree with the referee that the model data comparison indicate that the BE is not sufficient to reproduce the present day εNd isotopic signature in the Med Sea. This point will be clarified in the revised version (See change: P1-L11; P11-L10; P11-L24; P12-L25; P13-L3).

However the temporal and geographical variations of εNd could represent an interesting insight of Nd as tracer of the Mediterranean Sea circulation, in particular in the context of paleo-oceanographic applications.

P. 4, line 8 and throughout the text, as well as the figure caption. The compilation by Tachikawa et al. (2004) does not include Vance et al. (2004). This reference should be cited systematically for the data-model eNd comparison.
Done (See change: P4-L9; P10-L16, p11-L4; Fig.4, Fig.5).

P. 4, line 21, "extrapolate". For the eNd compilation of margins, I expect that both extrapolation and interpolation were applied.

Yes, referee is right; both extrapolation and interpolation were applied for each geological province. Clarified in the revised manuscript (See change P4-L27; P6-L25).

P.10, line 13, about correlation coefficient shown in Table 3. The coefficients shown in the text (0.71 and 0.61) do not correspond to the values indicated in Table 3. Please check and correct them.
Corrected (See change P10-L24).

P. 10, line 14, "dashed line" that is not shown in Fig. 4. I had already mentioned this point in my previous review. Please correct it.
We thank the referee for this suggestion; corrected in the revised manuscript. (See P10-L20).

P. 10, lines 15-16, "a slight overestimation of eNd between 0.3 and 1 eNd unites".
According to the new Fig. 4b, the size of the offset should be larger (ex. the offset seems to be -2 at around 200m). Please check.
The referee right, the offset should be larger for the intermediate/deep waters; this will be corrected in the revised manuscript (See P10-L22).

P. 10, line 18, "reasonable East-West gradient of eNd". What does "reasonable" mean? It is necessary to change this ambiguous expression.
Changed to:
The horizontal distribution of εNd (Appendix 2, 3 and 4) confirms this statistical correlation, showing that only EXP2 and EXP3 produces the most valid East-West gradient of εNd relative to in-situ data (See P10-L24).

P.11, lines 5-6. The eNd overestimation at the intermediate water depths is not limited for the Sicily Strait and Tyrrhenian sub-basins. The Alboran Sea presents a large offset (Fig. 6). Also "but the lack of observations prevent us to assess their consistency" is not appropriate and inconsistent with the authors answer ("the LIW layer gave an isotopic signature of almost -7 ± 1; Tachikawa et al., 2004; Henry et al., 2004, and from P. Montagnia, in prep.").
We thank the referee for this recommendation, you are right, the simulation present a large offset in the Alboran sub-basin. The unique available observation in the Alboran sub-basin is located close to the Gibraltar strait, and show εNd values characteristic of the outflow from the Atlantic sector. The model fail to reproduce this signal associated to the advection of water mass of Atlantic origin (see fig 5a), due to a simulated net water flux input from the Atlantic that stand in the lower range compared to observations (Beuvier et al, 2012).

Especially high εNd signatures are simulated in the Aegean sub-basin, over the Sicily channel and in the Tyrrhenian sub-basins (Fig. 5b).
This sentence has been changed for the sake of clarity, (See P11-L10; P13-L3).

P.11, line 15. Please clarify "any specific isotopic signature".
By using this sentences, we simply means that the station 51 (33.5 °N, 27°E) of in-situ data from Tachikawa et al., (2004) gave a relatively homogeneous Nd isotopic signature over the whole water column in the western Levantine basin.
This sentence has been changed for the sake of clarity, (See P11-L20).
station 51 (33.5 °N, 27°E) in the western Levantine basin expose a relatively homogeneous vertical isotopic signature that show no specific signal related to LIW.

Changed to: (See change P11-L24-27).

Except in the Alboran sub-basin, where a pronounced mismatches are simulated between model and observations, the model captures correctly the vertical profiles of Nd isotopic signatures especially in the Levantine sub-basin (averaged over the entire water column), especially producing a realistic and significant radiogenic signature associated to LIW at the Intermediate level (Fig.6), although the $\epsilon$Nd values can be overestimated in some places by almost 2 $\epsilon$Nd units (Fig.4).

Changed to: (See change P12-L25-28).

The high resolution simulation presented here provides a too radiogenic signature of Nd isotopic signature in the Med Sea, nevertheless this approach confirm the primordial role of the BE as the major source of Nd in the marine environment, Similar to what has been previously demonstrated for the global ocean (Arsouze et al., 2007) and the Atlantic basin (Arsouze et al., 2010).

Changed to: (See P13-L3).

especially in the Aegean and the Alboran sub-basins.

Tachikawa et al. (2004) have measured a very radiogenic value of -4.8 in the in the easternmost Levantine sub-basin at 200 m. the model simulate correctly this value, which proves the primordial role of Levantine sub-basin on the Nd isotopic distribution in the EMed ( as the Nile contribution is not included in this simulation). This radiogenic level observed and correctly simulated could impact the LIW signature (formed in the Levantine sub basin). The LIW represents the principal movement of water mass from the EMed into the WMed (Roether et al 2013). The model used in this study satisfactorily simulates the main structures of the thermohaline circulation of the Mediterranean Sea, with mechanisms having a realistic timescale compared to observations as evaluated in a previous study ( Ayache et al 2015a, Ayache et al 2015b; Pamieri et al 2015).

For the sake of clarity we change the cited section to (See P13-L20):

The LIW layer is particularly characterized by the most radiogenic signature in the intermediate level between 200 and 600 m, which is in good agreement with in-situ observations from Tachikawa et al. (2004) especially with the highest eNd value of -4.8 found at about 200 m in the easternmost Levantine basin. The LIW represents the principal movement of water mass from the EMed into the WMed.

This LIW signature is conserved in the WMed allowing us to study the impact of interannual variability, including the exceptional events observed in the ventilation of the deep waters (e.g., EMT) in the whole basin.

Figure 4. Label "a" and "b" is missing. On the x-axis of Fig. 4b, "epsilon" is not correctly shown.
Done

Figure 5. I am not sure that data from Henry et al. (1994) and Vance et al. (2004) are shown. Please check and correct, if necessary.
Done

Figure 6. I would put the Alboran result on the left, the Levantine result in the middle because of their east-west position in the Mediterranean Sea. "Tachikawa et al., 1983" should be "Tachikawa et al., 2004". Add also the other references (ex. Vance et al., 2004; Henry et al., 1994) when necessary.
Done, we thank the referee for the correction.

Figure 7a and 7b. The cyan curves show Levantine eNd variation, not "Aeg".
Done
* * *
Referee #2

Specific comments
After I read this manuscript, the following idea came to my mind: I wonder what is the aim of this study?
As written in the first sentence of discussion section, one important finding of this study is that the main features of the Nd IC distribution in the Mediterranean Sea are generated by assuming BE as the only Nd oceanic source term, which has been already demonstrated for the global ocean and the Atlantic basin. This fact confuses me a lot, because the authors have already found that their approach could be applied to much wider oceanic regions than the Mediterranean Sea. Furthermore, the Mediterranean Sea is a semi-closed basin and seems to be much easier system for modeling study than the global ocean. Therefore, I am not quite sure why they studied this oceanic region at this moment, which should have been done much earlier stage. I admit that more detailed and precise geological information might be available around the Mediterranean Sea than the global ocean, and this would lead to facilitate a more accurate simulation on Nd IC distribution. In reality, however, according to this manuscript, some problems (more radiogenic values in some areas etc.) still remain to be solved for a realistic simulation. Therefore, I recommend the authors to emphasize what is the merit to do modeling work for the Mediterranean Sea comparing with the global ocean. Otherwise, this paper would only deal with a case study of a limited oceanic region.
The processes leading to BE are still not fully understood yet (Jeandel and Oelkers, 2015). During the last few years, significant progress has been made in understanding how different water masses acquire their Nd Isotopic Composition (Tachikawa, 2003; Lacan and Jeandel, 2005). The modelling studies have reached the same conclusions on the relative importance of the BE on the Nd oceanic cycle on the global scale, although dust and river inputs could locally affect the surface waters (Arsouze et al., 2007; Siddall et al., 2008; Arsouze et al., 2009; Rempfer et al., 2011).

In this context this study presents a new high resolution map of Nd IC of all the Med Sea margins, this map allow a significant improvement compared to the global maps of Jeandel et al (2006), and give a more realistic representation of this very complex morphology and geology in the Mediterranean region. Starting from this maps we have made a simulation with the BE as the unique Nd source following Arsouze et al., (2007) approach in order to estimate the relative importance of this source compared to the other source in this semi enclosed basin.

The model data comparison indicate that the BE is not sufficient to reproduce the present day eNd isotopic signature in the Med Sea, however this source is the dominate Nd input in the Med Sea (See change P4-L7-14).

This study is part of the work carried out to assess the robustness of the NEMO-MED12 model, used to study the thermohaline circulation and the biogeochemical cycles in the Mediterranean Sea, to improve our ability to predict the future evolution of this basin under the increasing anthropogenic pressure.

For the sake of clarity section was modified to: (See change P4-L7-17).

Arsouze et al., (2007) approach to simulate the BE was evaluated and largely accepted by the scientific community (e.g. Arsouze et al 2009), following this protocol made up for the global scale, we implemented the neodymium in a high-resolution regional model (NEMO-MED12) developed for the Med Sea. We used dissolved $\varepsilon$Nd data compiled by Tachikawa et al. (2004), Vance et al., (2004), and Henry et al., (1994) to evaluate the ability of this model to reproduce the main features of the circulation and mixing of the Med Sea water masses for which Nd signatures are known. These tools provided perspectives on i) the $\varepsilon$Nd distribution in the whole Med Sea, ii) the impact of the inter-annual variability of the thermohaline circulation (e.g., EMT event) on the modelled $\varepsilon$Nd distribution, and iii) high resolution of the geological field on the one hand and the model on the other hand allows revealing possible local heterogeneities which could reflect local BE effects.

Technical corrections

P4 L28; What is "rive"? Please correct.
Replaced by coastline (See change P5-L2).

P7 L28; "(Ludwig et al., 2009)" should be "Ludwig et al. (2009)".
Done (See change P7-L24).

P14 L15; "Our next step is" seems to be much proper than "Our next step was".
Done (See change P14-L28).

Figs 4 and 5; Although the authors show new figures in the documents for replying referees' comments, those figures presented in the revised manuscript are the previous ones. Please replace to the new ones.
Done

**Lastly, we thank the anonymous reviewer again for the helpful comments towards improving the manuscript**

**References**

[revised manuscript text omitted]